



# An aerosol vertical data assimilation system (NAQPMS-PDAF v1.0): development and application

Haibo Wang[1,4][★], Ting Yang[1,3*][★], Zifa Wang[1,3,4*], Jianjun Li[2], Wenxuan Chai[2], Guigang Tang[2], Lei Kong[1,4], Xueshun Chen[1,3]

[1] The State Key Laboratory of Atmospheric Boundary Layer Physics and Atmospheric Chemistry, Institute of Atmospheric Physics, Chinese Academy of Sciences, Beijing 100029, China

[2] China national Environmental Monitoring Centre, Beijing

[3] Center for Excellence in Regional Atmospheric Environment, Institute of Urban Environment, Chinese Academy of Sciences, Xiamen 361021, China

[4] College of Earth and Planetary Science, University of Chinese Academy of Sciences, Beijing 100029, China
[★] These authors contributed equally to this work

*Corresponding author: Ting Yang (tingyang@mail.iap.ac.cn) , Zifa Wang (zifawang@mail.iap.ac.cn)

**Abstract.** Aerosol vertical stratification is important for global climate and planetary boundary layer (PBL) stability, and no single method can obtain spatiotemporally continuous vertical profiles. This paper develops an online data assimilation (DA) framework for the Eulerian atmospheric chemistry-transport model (CTM) Nested Air Quality Prediction Model System (NAQPMS) with the Parallel Data Assimilation Framework (PDAF) as the NAQPMS-PDAF for the first time. Online coupling occurs based on a memory-based way with two-level parallelization, and the arrangement of state vectors during the filter is specifically designed. Scaling tests demonstrate that the NAQPMS-PDAF can make efficient use of parallel computational resources for up to 2.5 k processors with weak scaling efficiency up to 0.7. One-month-long aerosol extinction coefficient profiles measured by the ground-based lidar and the concurrent hourly surface $PM_{2.5}$ are solely and simultaneously assimilated to investigate the performance and application of the DA system. The hourly analysis and subsequent one-hour simulation are validated through lidar and surface $PM_{2.5}$ measurements assimilated and not assimilated. The results show that lidar DA can significantly improve the underestimation of aerosol loading, especially at a height of approximately 400 m in the free-running (FR) experiment, with the BIAS changing from -0.20 (-0.14) 1/km to -0.02 (-0.01) 1/km and correlation coefficients increasing from 0.33 (0.28) to 0.91 (0.53) averaged over sites with measurements assimilated (not assimilated). Compared with the FR experiment, simultaneously assimilating $PM_{2.5}$ and lidar can have a more consistent pattern of aerosol vertical profiles with a combination of surface $PM_{2.5}$ and lidar, independent extinction coefficients from the Cloud-Aerosol Lidar with Orthogonal Polarization (CALIOP), and aerosol optical depth (AOD) from the Aerosol Robotic Network (AERONET). Lidar DA has a larger temporal impact than that in $PM_{2.5}$ DA but has deficiencies in subsequent quantification on the surface $PM_{2.5}$. The proposed NAQPMS-PDAF has great potential for further research on the impact of aerosol vertical distribution.





# 1 Introduction

Aerosol vertical distribution has a significant impact on the estimation of the global budget of aerosols on climate (Torres et al., 1998; Peters et al., 2011; Meyer et al., 2013), planetary boundary layer (PBL) stability (Li et al., 2017b, 2020b; Su et al., 2020), the understanding of the aerosol evolutionary process and surface concentration (Chen et al., 2009; Liu et al., 2018;

Quan et al., 2020) and the retrieval of aerosol optical properties from passive sensors (Li et al., 2020a).

In a broad sense, aerosol optical depth (AOD) measurements, which are the vertical integral of aerosol extinction coefficients, can be deemed to include vertical information and have relatively low uncertainty (Holben et al., 2001). However, AOD with passive remote sensors can only be used to investigate the qualitative impact of aerosol vertical distribution (Zhu et al., 2018) and quantitative relationship with surface concentration, which neglects vertical information to some extent (Li et al., 2018b;

Yang et al., 2019; Wei et al., 2021). Aircraft measurements can directly provide aerosol vertical mass concentration profiles (Bahreini, 2003; Chen et al., 2009; Liu et al., 2019) but are limited in spatiotemporal coverage due to expensive costs at present. Lidar, an effective tool to measure aerosol stratification with active remote sensors, is widely used in vertical research on aerosols (Shimizu, 2004; Liu et al., 2013; Sicard et al., 2015; Proestakis et al., 2019; Mehta et al., 2021) and is generally composed of ground-based and space-borne lidar. Compared with the relatively large retrieval errors and sparse coverage from

satellite observations, ground-based lidar measurements are more accurate (Cheng et al., 2019a). However, although ground-based lidar generally has more intensive coverage than space-borne measurements, it can only provide single-point information with limited spatial coverage. The three-dimensional structure of aerosols, especially their vertical structure (Solazzo et al., 2013; Kipling et al., 2016), can be simulated by the atmospheric chemistry-transport model (CTM), which nonetheless has large uncertainties in chemical initial/boundary conditions, meteorological initial/boundary conditions, emissions, and

parameterizations of physical and chemical processes (Wu et al., 2020b) and may differ substantially from the real situation.

Currently, there is no single method to obtain spatiotemporally continuous data of aerosol vertical profiles. Therefore, a combination of the advantages of these two approaches can provide more accurate aerosol profiles, which can be done through data assimilation (DA). DA is a technique that combines observations of a system, including their uncertainty, with estimates of that system from a numerical model, including its uncertainty, to obtain a more accurate description of the system including

an uncertainty estimate of that description (Evensen, 2009; Vetra-Carvalho et al., 2018). Currently, ensemble Kalman filter (EnKF) (Evensen, 1994)-based methods and three-dimensional/four-dimensional variational methods (3D/4D-Var) (Schlatter, 2000) are mainstream algorithms. Variational methods are restricted in adequately quantifying the flow-dependent background error and have to program complex adjoint operators (Bannister, 2017). The EnKF, originating from the merger of the traditional Kalman filter (KF) theory (Kalman and Bucy, 1961) and Monte Carlo estimation methods, uses an ensemble of

possible realizations containing valuable flow-independent information (Houtekamer and Zhang, 2016) and has a variety of variants, such as the ensemble square root filter (EnSRF), the local ensemble transform Kalman filter (LETKF) (Bishop and Toth, 1999; Hunt et al., 2007) and the local error-subspace transform Kalman filter (LESTKF) (Nerger et al., 2012). DA has been used in meteorology and oceanography to improve forecasts and construct reanalysis for many decades (Park and Xu,



2009; Lahoz et al., 2010). The application of DA in atmospheric chemistry has only occurred since the mid-1990s (Bocquet et

al., 2015) and has mainly involved producing accurate air pollutant concentration analyses and forecasts (Tang et al., 2011; Peng et al., 2017; Ma et al., 2019), improving the inversion of emissions (Tang et al., 2016; Kong et al., 2019; Feng et al., 2020; Wu et al., 2020a), inverting model parameters (Bocquet, 2012) and constructing air quality reanalysis datasets (Lynch et al., 2016; Kong et al., 2020). These common studies mentioned above have mainly concentrated on the performance of assimilating surface measurements in CTMs.

As lidar is the most powerful instrument to measure aerosol vertical information, lidar DA with CTMs has naturally become a popular research area. The study on lidar DA started in 2007, as optimized dust emissions were obtained by assimilating National Institute for Environmental Studies (NIES) lidar measurements during the extreme dust phenomenon on 30 April 2005 (Yumimoto et al., 2007). Since then, a few studies have been conducted on the application of lidar DA: studies on assimilating virtual lidar measurements based on Observing System Simulation Experiment (OSSE) (Wang et al., 2013),

studies on assimilating space-borne Cloud-Aerosol Lidar with Orthogonal Polarization (CALIOP) (Sekiyama et al., 2010; Zhang et al., 2011; Cheng et al., 2019b), studies on investigating the short-term (no more than 12 hours) performance of analyses and subsequent forecasts (Wang et al., 2014b, a; Cheng et al., 2019a; Liang et al., 2020), studies based on static background DA methods (Wang et al., 2013, 2014b, a; Zheng, 2018; Xiang, 2018; Cheng et al., 2019a; Liang et al., 2020) and studies on ground-based lidar concentrated on a few model grids (Ma et al., 2020).

To conduct ensemble DA research, especially for observations including vertical profile information, an applicable DA framework is strongly needed. Open source generic DA frameworks have recently been developed to further promote the development of DA, as well as frameworks for specific Earth system components, such as the Weather Research and Forecasting model Data Assimilation system (WRFDA) (Barker et al., 2012). The Gridpoint Statistical Interpolation (GSI) is an operational DA system developed by the National Centers for Environmental Prediction (NCEP) Environmental Modeling

Center (EMC) (Wu et al., 2002). GSI supports a variety of conventional observations with a specific BUFR format, as well as satellite radiance/brightness observations that are based on the especially developed Community Radiative Transfer Model (CRTM) (Liu and Lu, 2016). However, the implementation and application of ensemble DA is not the key point of GSI, although GSI supports classic EnKF and hybrid DA. OpenDA (http://www.openda.org/, last access: 9 September 2021) is an open source toolbox for the development and application of DA algorithms, which is written in JAVA and some parts in C

(van Velzen et al., 2016). While OpenDA only contains classic DA methods such as 3DVar, EnKF and EnSRF, it is mainly used in hydrodynamic models (Ridler et al., 2014; Garcia et al., 2015; van Velzen et al., 2016; Baracchini et al., 2020) and has a tentative application on indoor contaminant concentrations (Lin and Wang, 2013). The Data Assimilation Research Testbed (DART), developed by the National Center for Atmospheric Research (NCAR), is an open source community facility for ensemble DA (Anderson et al., 2009). The DART has been coupled with the Weather Research Forecasting Model with

chemistry (WRF-CHEM) (Mizzi et al., 2016, 2018) and is utilized to study the impact of assimilating air pollutant measurements into chemistry transport models (CTMs) (Ma et al., 2019, 2020; Emili et al., 2019; Zhang et al., 2021). The Parallel Data Assimilation Framework (PDAF), a community open source project and software environment for ensemble DA,



is dedicated to simplifying the implementation of coupling the DA framework with existing numerical models (Nerger and Hiller, 2013). The PDAF can provide fully implemented and optimized filter algorithms that are model agnostic, such as the LETKF and LESTKF with standardized interfaces. The PDAF is widely applied in numerical models of the atmosphere, land surface and ocean (Kurtz et al., 2016; Chen et al., 2017; Yu et al., 2018; Gebler et al., 2019; Gillet-Chaulet, 2020; Tang et al., 2020; Stepanov et al., 2021) but has not yet been coupled with the CTM for the study of air quality and further assimilation of vertical profile measurements.

In this study, we present a DA system coupling the Nested Air Quality Prediction Model System (NAQPMS) (Li et al., 2012; Wang et al., 2014c) with the PDAF as the NAQPMS-PDAF with good expandability. To the authors' knowledge, this is the first attempt to couple the PDAF with the CTM. The coupling is performed online using memory-based communication, and the detailed implementation is discussed. Afterward, one-month ground-based lidar and surface $PM_{2.5}$ measurements are assimilated into the NAQPMS-PDAF to preliminarily evaluate the performance of this DA system. The vertical distribution of aerosol analysis and subsequent one-hour simulation, which can be regarded as a one-month reanalysis dataset, are investigated with the application of the NAQPMS-PDAF. The remainder of this study is structured as follows. Section 2 introduces the NAQPMS and PDAF and describes the technical implementation of the NAQPMS-PDAF as well as the ensemble filter algorithm used in this study. The model configuration and observation data as well as the experimental setting are also discussed in Section 2. Section 3 presents the results and discussion, mainly including the scaling behavior, the evaluation of the NAQPMS-PDAF with internal check and independent validation, and the analysis of aerosol vertical profiles and uncertainties of the NAQPMS-PDAF. The conclusions and outlook are finally summarized in Section 4.

## 2 Methodology and data

### 2.1 Regional chemical transport model NAQPMS

The NAQPMS, a three-dimensional CTM, was developed by the Institute of Atmospheric Physics (IAP), Chinese Academy of Sciences (CAS) and employed in this paper. The NAQPMS uses a multiscale domain-nesting technique to treat physical and chemical processes of aerosols and gases on global and regional scales. The model calculates multi-dimensional turbulent diffusion (Byun and Dennis, 1995) and advection process (Walcek and Aleksic, 1998). The Carbon Bond Mechanism (CBM-Z), including 71 species and 134 reactions, was used to represent gas-phase chemistry (Zaveri and Peters, 1999). The NAQPMS contains dry deposition processes at the surface (Zhang et al., 2003). The model's aqueous chemistry and wet scavenging are adapted from the Regional Acid Deposition Model's second generation (RADM2) (Stockwell et al., 1990). The composition and phase state of an $NH_4^+ - SO_4^{2-} - NO_3^- - Cl^- - Na^+ - H_2O$ inorganic aerosol system is calculated by the thermodynamic model ISORROPIA (Nenes et al., 1998), and secondary organic aerosols are calculated based on the volatility basis set (VBS) (Donahue et al., 2006). Heterogeneous reactions involving ozone, sulfate, soot, dust and sea salt and an accurate radiative transfer model (TUV, version 4.5) (Li et al., 2011) are included to simulate the mixing process between aerosols and gaseous



pollutants (Li et al., 2018a). The online emissions of dust (Wang et al., 2000), dimethyl sulfide (Lana et al., 2011) and sea salt
(Athanasopoulou et al., 2008) are calculated by the NAQPMS. The Weather and Forecasting model (WRF), driven by Final
Analysis (FNL) data from the National Centers for Environmental Prediction (NCEP), provides the meteorology field.

One of the optical modules in the NAQPMS to simulate the extinction coefficients used in this study is a "reconstructed
extinction coefficient" method proposed by Malm (2000) as a part of the Interagency Monitoring of Projected Visual
Environment (IMPROVE) program. The IMPROVE method is widely used (Pitchford et al., 2007; Shen et al., 2014; Bai et
al., 2020) as a linear equation and has advantages in computational complexity compared with the radiative transfer equation
and Mie equation. The IMPROVE equation used in the NAQPMS has been widely applied in simulating aerosol optical
properties, which can effectively reproduce the aerosol vertical distribution on the North China Plain (NCP) (Li et al., 2012,
2014, 2017a; Wang et al., 2017b). The difference between aerosol optical properties at 532 nm (lidar measurement) and 550
nm (IMPROVE method) is neglected in this study.

**2.2 PDAF**

The PDAF (http://pdaf.awi.de, last access: 2 October 2021) is a community open-source project that was committed to
facilitate the implementation and application of DA algorithms with large-scale numerical models (Nerger and Hiller, 2013).
A generic framework provided by the PDAF contains fully integrated, parallelized and optimized ensemble assimilation
methods such as classical EnKF (Evensen, 1994; Burgers, 1998), LESTKF (Nerger et al., 2012), LETKF (Hunt et al., 2007)
and LNETF (Tödter and Ahrens, 2015) and smoother algorithms (Nerger et al., 2014). Furthermore, the PDAF provides
template routines called as call-back routines to interface the DA system with the numerical model and it provides strategies
for establishing parallel communication for DA algorithms and ensemble simulations that are model agnositic. The
standardized interfaces provided by the PDAF can allow the further development of the numerical model and assimilation
methods independently.

The call-back routines for users can be divided into four different types, which can be seen in Fig. 4 in Nerger et al.
(2020). The first kind of routine is utilized to interface model fields from the numerical model with the state vector in the
PDAF before and after each assimilation cycle, such as the routines *collect_state_pdaf* and *distribute_state_pdaf*. The second
kind of routine refers to observation handling mainly used to map the state vectors into the observed state vectors and set
corresponding measurement uncertainties. Localization is the third type and is discussed in Section 2.3.3 and Section 2.3.1.
The last kind of routine is for pre- and postprocessing of the ensemble members in the DA system, which mainly produces
ensemble means for the analysis and forecasting.

The data coupling between the numerical model and DA system can be divided into two ways (offline coupling and online
coupling) and can be alternatively implemented in the PDAF. The offline method exchanges data principally through the
input/output (I/O) files from the numerical model and DA system, while the online method occurs through main memory.
Specifically, the model output after one step of ensemble simulations is written to files in offline coupling, which is the
subsequent input of the DA system. When the DA system of the NAQPMS-PDAF performs the filter algorithm, its output,





called as restart files, is written to the hard disk, which is the initial condition of the next step of the numerical model. The offline coupling is fit for the source code of the numerical model which cannot be modified. It inevitably produces an excess I/O overhead with too much extra storage loading in offline coupling, which tremendously increases the total running time

because each ensemble member is needed for the computation of the analysis and forecast in the DA system. In online coupling, the numerical model only needs to be initialized once with a lower overhead. The entire coupled DA system has a better computational efficiency without frequent I/O operation compared with offline coupling. One drawback of online coupling is that it needs more programming effort than offline coupling (Gropp et al., 1994). The PDAF can be integrated into the numerical model with simplified implementation based on the source code, which is exemplified using the

NAQPMS in Section 2.3.1. It avoids an indirect data transfer with a variety of I/O files between the DA system provided by the PDAF and the numerical model.

## 2.3 NAQPMS-PDAF

### 2.3.1 Technical implementation

As the NAQPMS described in Section 2.1 is well written and its source code is available, this study chooses the online method

to couple the PDAF with the NAQPMS in order to gain the best performance. The core modification in the coupling is parallelization for ensemble simulations. The NAQPMS is parallelized utilizing the Message Passing Interface standard (MPI; Gropp et al., 1994), which is also used in the PDAF in Fortran. MPI allows each process to handle distributed parts of a program and data exchange. The parallel communication among groups of processes is called the communicator in MPI. Generally, the communicator MPI_COMM_WORLD is used in large-scale numerical models to split the whole program into

several processes. The communicator MPI_COMM_WORLD is also used in the NAQPMS as one-level parallelization to improve computational efficiency. The distribution of processes in the NAQPMS is exemplified in Fig. 1a with the global communicator MPI_COMM_WORLD. The three-dimensional model domain is split into four columnar regions (four processors in the communicator MPI_COMM_WORLD) with a division of processors in a Cartesian grid, which are run in four processors. However, the one-level parallelization in the NAQPMS can only satisfy the need of one model realization. To

perform ensemble simulations and filters in the DA framework, it is necessary to introduce one additional level of parallelization. Therefore, the communicator MPI_COMM_WORLD is split into three other communicators which are the model communicator (MPI_COMM_MODEL), the filter communicator (MPI_COMM_FILTER) and the coupling communicator (MPI_COMM_COUPLE) on the routine *init_parallel_pdaf* in the PDAF.

The apportionment of the processes with three communicators is shown in Fig. 1b-d with four processes within one model

realization and a total of three ensemble members in the NAQPMS-PDAF. The number of model communicators is equal to that of ensemble members, with four processors within one model communicator in Fig. 1b. This indicates that the three ensemble members are running simultaneously with different initial perturbations. The main modification in the source code of the NAQPMS is to replace MPI_COMM_WORLD with MPI_COMM_MODEL and add routines concerning the PDAF.





Figure 2 shows the processor layout of four communicators with the same configuration in Figure 1. The filter communicator
performs the core DA algorithm with four processors, the number of which is equal to that of processors in one model
realization. In the filter communicator, only the processes in one ensemble member (four orange processors in Fig. 2) are used,
and other processes keep idle during the ensemble DA. The couple communicator with four tasks (three processors in one task)
is used to exchange information between the processes in the model and filter communicator before and after the DA step. The
number of processors in one couple communicator is equal to that of the ensemble members. This indicates that one task in
the couple communicator focuses on the same subdomain for collecting the information in ensemble simulations and the filter
algorithm. The different decompositions between the model communicator and filter communicator as well as the couple
communicator (Fig. 1b-d) are explained below.

The main program flow of the NAQPMS-PDAF in this study can be generalized in the following steps:

1. initialization of MPI (MPI_COMM_WORLD);

2*. initialization of three communicators (MPI_COMM_MODEL, MPI_COMM_FILTER and MPI_COMM_COUPLE);

3. initialization for the NAQPMS;

4*. initialization of variables for the PDAF;

5. time loop with ensemble simulations and filter:

a*. update the model variables corresponding to state vectors in the NAQPMS;

b. advance the NAQPMS to the next assimilation time step;

c*. update the state vector after ensemble integration;

d*. filter step;

6*. finalization of the PDAF and NAQPMS.

The program flow shown above is based on the original flow of the NAQPMS, while the steps denoted by asterisks are
additionally added by the PDAF due to the online coupling. After the core modification of 2-level parallelization discussed
above, three communicators are initialized in steps 1 and 2 at the very beginning. The model initialization of the NAQPMS
proceeds in step 3. Each processor in the communicator MPI_COMM_MODEL opens the same common input file, such as
meteorological data model configuration files, and a different input file of initial perturbed emission for the ensemble
initialization (see Section 2.3.3). One processor within one model realization (communicator MPI_COMM_MODEL) reads
different parts of the opened data file corresponding to the different subdomains assigned in steps 1 and 2.

In step 4, the variables (size of state vectors, domain setting in the DA framework and so on) in the PDAF are initialized.
The dimension of the state vectors used in the DA framework is redesigned to meet the needs of domain decomposition and
the observation operator in the PDAF. As shown in Fig. 1e, ix, iy and iz are the number of grids along the longitudinal,
latitudinal and vertical directions, respectively. S(.) represents the specific state variables, and the number of variables is N.
Specifically, the dimensional order from outside to inside of the state vector is ix, iy, and iz variables. As shown in Fig. 1c, the
domain decomposition follows the longitudinal direction. This indicates that regardless of the position to be cut in the
longitudinal direction, the slice has full three-dimensional information with all variables that can map into the observation.





The observation vectors have the same configuration, while they have only one variable. The time loop of model simulations and filter update in the NAQPMS-PDAF occurs in step 5. Step 5b represents the evolution of air pollutants under atmospheric

physical and chemical processes, which follows the raw source code of the NAQPMS. The ensemble filter algorithm is implemented after advancing the NAQPMS at the end of each time loop of ensemble simulations. The number of time steps is set in the initialization of the NAQPMS, and the number of filter steps is set in that of the PDAF, while these two terms are not necessarily the same. Before and after the time loop of the NAQPMS, two modules, mainly including routines *from_field* and *into_field,* are run (step 5a and step 5c). The routine *from_field* collects all model state vectors from each processor on one

model communicator (MPI_COMM_MODEL), which is updated by the filter period at the previous cycle. Afterward, the routine *from_field* apportions the state vectors into subvariables. The routine *into_field* does the reverse similar to that in the routine *from_field*. After the performance of the routine *into_field*, each process gains the state vectors by cutting along the longitudinal direction. The domain decomposition in the model and couple communicator, as well as the filter communicator, is different which are discussed in step 4, which seems to be a waste of computational efficiency. However, the vector states

used in this study (see Section 2.3.3) are only a rare part of all variables in the NAQPMS. Hence, it has little impact on the DA framework, and the parallel efficiency is high, which is discussed in Section 3.1. In step 5d, the ensemble filter is called to update the state vectors with observations, which are implemented in the source code of the PDAF, which may not need to change unless a new filter algorithm is added. In step 6, the variables used in the NAQPMS and PDAF are deallocated, and timing information and memory information are provided that can check the program run and help to optimize the program.

In practice, we developed a pre/postprocessing auxiliary program written in Python 3.6 for the task of constructing a three-dimensional data structure of observation and plotting figures for different aims. We also set a variable to identify each run in the NAQPMS-PDAF and to create new files corresponding to that variable to save the data to avoid frequently removing and overwriting the data that we need.

### 2.3.2 Ensemble DA algorithm

The error subspace transform Kalman filter (ESTKF) is used in this study, which is outlined in this section. As an EnKF-based method, the algorithm can be decomposed into the steps of forecasting and analysis. The forecast propagates the states and the corresponding error covariance matrix forward in time from a previous analysis at $t = t_{k-1}$ to the next observation time $t = t_k$. The numerical model $M_k$, the NAQPMS in this study, is used to propagate ensemble members during certain time steps:

$$x_i^{f,k} = M_k(x_i^{a,k-1}), \tag{1}$$

where $x_i^{f,k}$ is the model forecast realization at $t = t_k$ and $x_i^{a,k-1}$ is the model analysis realization at $t = t_{k-1}$. $N_e$ is the number of ensemble members and $i = 1, 2,..., N_e$. The size of state vector ($x_i^{f/a}$) is $N_x$. The state vector in the forecast phase can be estimated and is approximated by the ensemble mean:





$$\overline{x}^{f,k} = \frac{1}{N_e}\sum_{i=1}^{N_e} x_i^{f,k}. \tag{2}$$

In the analysis step at $t = t_k$, the filter algorithm transforms a matrix of forecast ensemble $X_k^f$ into a matrix of analysis

ensemble $X_k^a$, and the time index k is omitted because the time of all the implementations during the analysis step is constant.

$$X^a = \overline{x}^f 1_{Ne}^T + X^f(\omega 1_{Ne}^T + \tilde{W}), \tag{3}$$

where $X^a$ and $X^f$ denote an ensemble matrix in which each of the $N_e$ columns represents one model realization of the

analysis and forecast, respectively ( $X^{a/f} = (x_1^{a/f}, x_2^{a/f}, ... x_{N_e}^{a/f}) \in R^{N_x \times N_e}$ ). $1_{Ne}^T$ is a identify matrix of size $N_e \times 1$. Furthermore,

$\omega$ transforms the ensemble mean with size $N_e$ and $\tilde{W}$ transforms the ensemble perturbations with size $N_e \times N_e$. The sample

error covariance matrix approximated with an ensemble is only a low-rank approximation of the true state, and its rank is at

most $N_e - 1$ (Gillet-Chaulet, 2020). This property is utilized in the ESTKF to compute the ensemble matrix transformed in

this subspace (Nerger et al., 2020). The error-subspace matrix is computed by

$$L = X^f \Omega, \tag{4}$$

where it is a projection matrix of size $N_e \times (N_e - 1)$ given by the set of equations as follows:

$$\Omega_{ij} = \begin{cases} 1 - \dfrac{1}{N_e}\dfrac{1}{\dfrac{1}{\sqrt{N_e}}+1} & \text{for } i = j, i < N_e \\[4mm] -\dfrac{1}{N_e}\dfrac{1}{\dfrac{1}{\sqrt{N_e}}+1} & \text{for } i \neq j, i < N_e \\[4mm] -\dfrac{1}{\sqrt{N_e}} & \text{for } i = N_e. \end{cases} \tag{5}$$

The state vectors are mapped into the observation space with the observation operator H in the analysis step by

$$y = H(x^f) + \varepsilon, \tag{6}$$

where $\varepsilon$ is the observation error assumed to be an unbiased Gaussian distribution and it has known observation error

covariance matrix $R$. In the error subspace of the ensemble DA, an ensemble transform matrix is computed especially with

Eqs. (5) as

$$A^{-1} = \rho(N_e - 1)I + (HX^f\Omega)^T R^{-1}HX^f\Omega, \tag{7}$$

where the factor $\rho$ with $0 < \rho \leq 1$ is defined as the forgetting factor. The forecast error covariance matrix is inflated by the

forgetting factor $\rho$ to overcome the undersampling issues (Anderson and Anderson, 1999; Tuan Pham et al., 1998).

Thus, the weight vector $\omega$ and matrix $\tilde{W}$ are given by

$$\omega = \Omega A(HX^f\Omega)^T R^{-1}(y - H\overline{x}^f), \tag{8}$$





$$\tilde{W} = \sqrt{N_e - 1}\,\Omega A^{1/2}\Omega^T, \tag{9}$$

where $A^{1/2}$ is the symmetric square root computed from the eigenvalue decomposition. Apart from the inflation discussed above, localization technology is also utilized for filter stabilization. Only observations within a certain horizontal localization radius are used when updating a local analysis. In addition, each observation is weighted with distance, which is performed by

adjusting the Eqs. (7) and (8). A fifth-order polynomial function with compact support (Gaspari and Cohn, 1999) is used to compute the weight of the horizontal localization. We set the horizontal localization radius to 200 km according to other studies (Kong et al., 2020; Zhao et al., 2020). Three adjacent grids are considered with the influence of observations of the vertical localization, which is a simple setting.

### 2.3.3 Model configuration

To reduce the uncertainty of meteorological data, three nested model domains for the WRF are performed with horizontal resolutions of 45, 15, and 5 km, as shown in Fig. 3a. The vertical coordinate system consists of 40 terrain-following levels with 27 layers within 2 km, which is especially designed for research on assimilating measurements, including vertical information. The third domain of the WRF is used as the only model domain in ensemble simulations in the NAQPMS-PDAF for saving computing resources and avoiding the error from performing multi-DAs for different domains. The WRF is driven

by the National Centers for Environmental Prediction (NCEP) Final Analysis (FNL) data. The lateral and upper boundary conditions were taken from the global chemistry transport model MOZART-v2.4 with a 2.8° horizontal spatial resolution (Brasseur et al., 1998; Hauglustaine et al., 1998). Anthropogenic emissions were provided by the 0.25° Multi-resolution Emissions Inventory for China (MEIC, http://www.meicmodel.org, last access: 14 September 2021).

     The state vectors include the mass concentration of ammonium, sulfate, nitrate, BC, OC, soil particulate matter in $PM_{2.5}$,

soil particulate matter in $PM_{10}$, sea salt, fine dust, coarse dust and RH (relative humidity). RH, one of the state vectors, is used to map the model space into the observation space. The state vectors are all from the IMPROVE equation. The ensemble of initial chemical conditions is generated by perturbing the emissions based on their error probability distribution functions (PDFs). The PDFs are assumed to be Gaussian distributions, and the uncertainty of each species follows the evaluation in Streets et al. (2003) and Zhang et al. (2006) (12 % for $SO_2$, 31 % for $NO_x$, 68 % for VOC, 53 % for $NH_3$, 70 % for CO, 132 %

for $PM_{10}$, 130 % for $PM_{2.5}$, 208 % for BC and 258 % for OC). In practice, a set of perturbation factors is implemented with the Schur product to the original emissions. An isotropic correlation model is utilized to impose a horizontal correlation of disturbed emissions. Following Kong et al. (2020), the decorrelation length is specified as 150 km in this study. Twenty smooth pseudorandom perturbation fields of perturbation factors are generated for each species. The observation error of the surface $PM_{2.5}$ measurement is set as 8 μg/m³ to account for measurement and representativeness (Ma et al., 2019). However, the

observation error of ground-based lidar measurements is set as 10 % of the mean aerosol extinctions in reference to other studies on lidar DA (Sekiyama et al., 2010; Cheng et al., 2019b; Ma et al., 2020) due to a lack of specialized study of the error of lidar measurements.



## 2.4 Observational data

The extinction coefficients used in this study are ground-based lidar measurements from 11 sites. As shown in Figure 3b, 11
lidar sites (red and blue dots) are distributed within the NCP with serial numbers from 1 to 11. The ground-based lidar from
No. 1 to No. 6 with AGHJ-I-LIDAR (HPL) (Fan et al., 2019; Shi et al., 2020; Wang et al., 2020a) is distributed south of the
NCP, while the lidar from No. 7 to No. 11 with LGJ-01 (Sun et al., 2019) is located north of the NCP. These two lidar systems
have similar specifications, with an energy of approximately 25 mJ and a pulse repetition rate of 20 Hz. The tunable time
resolution is from 1 to 5 min, and the vertical resolution is 15 m. The blind ozone height is 300 m for both lidar types. Hourly
extinction coefficients profiles at 532 nm which can represent vertical distribution of aerosols are averaged from the raw lidar
measurements to improve the signal-to-noise ratio and meet the time resolution of the numerical model. To meet the model
domain, 40 layers are generated from the raw lidar data by a piecewise cubic interpolating Bezier polynomial (Alfeld, 1984)
using SciPy (Virtanen et al., 2020). As shown in Fig. 3b, mass concentration observations of surface $PM_{2.5}$ (green and pink
dots) on the NCP are obtained from the China National Environmental Monitoring Centre (CNEMC) (http://www.cnemc.cn/,
last access: 20 May 2021).

The Cloud-Aerosol Lidar and Infrared Pathfinder Satellite Observation (CALIPSO) mission (http://www-
calipso.larc.nasa.gov/) is the first satellite-borne lidar specifically designed for aerosol and cloud study and is a collaborative
effort between the NASA Langley Research Center (LaRC), Centre National d'Etudes Spatiales (CNES), Hampton University
(HU), Institut Pierre-Simon Lapace (IPSL), and Ball Aerospace and Technologies Corporation (BATC) (Winker et al., 2009).
CALIOP is an elastic backscatter lidar that is onboard CALIPSO. Vertical profiles of aerosol and cloud backscatter coefficients
at 532 nm and 1064 nm, as well as depolarization ratio profiles at 532 nm from -0.5 to 30 km, are available from CALIOP.
CALIOP has a horizontal spatial resolution of 5 km and a vertical spatial resolution of 40 m. The extinction coefficients at 532
nm from the CALIOP level 2 version 4.20 are used in this study for independent validation of the DA experiments. Fifty-two
CALIOP orbits are included within the model domain for April 2019.

The Aerosol Robotic Network (AERONET, https://aeronet.gsfc.nasa.gov/, last access: 9 Sep 2021) is a global network of
autonomously CIMEL sun photometer measurements that can provide high-quality key aerosol optical parameters at eight
spectral bands within a range from 340 to 1020 nm (Holben et al., 1998). AOD data from the AERONET Version 3 Level 2.0
database, which has assured quality after the screening of clouds, is used as the independent measurement to evaluate the DA
performance. AOD data at 532 nm, used as independent validation dataset, is calculated from the AERONET AODs at 440
nm, 500 nm and 670 nm by quadratic polynomial interpolation (Eck et al., 1999; Wang et al., 2020b).

## 2.5 Experimental setting and evaluation method

To investigate the performance of the NAQPMS-PDAF with real observations and the impact of assimilating vertical
observations into the CTM, a total of four experiments with a 30-day study period running from 00:00 UTC 1 April 2019 to
18:00 UTC 30 April 2019 were conducted. As we care more about the variables related to extinction coefficients and surface





PM$_{2.5}$, which exclude most gaseous pollutants, the number of state variables used and output in the assimilation part of the NAQPMS-PDAF was smaller than those in the NAQPMS. We ran the NAQPMS for 3 days from 00:00 UTC 29 March 2019 to 23:00 UTC 31 March 2019 as a spin-up for NAQPMS-PDAF. As summarized in Table 1, the first experiment of the NAQPMS-PDAF was free running (FR) without assimilating any observations, providing a comparison with the following DA experiments. An experiment assimilating the surface PM$_{2.5}$ (NP-PM25) was performed to investigate the performance of

the NAQPMS-PDAF. An experiment assimilating only the extinction coefficients measured by ground-based lidar (NP-LIDAR) was designed to examine the impacts of assimilating vertical observations of aerosols. An experiment simultaneously assimilating lidar measurements and surface PM$_{2.5}$ simultaneously (NP-LIDAR-PM25) was conducted to probe the combined impacts of the fusion of multiple observations containing surface and vertical observations.

During the evaluation, four basic statistical metrics, the root mean square error (RMSE), the mean absolute error (MAE),

the mean bias (BIAS) and the Pearson correlation coefficient (CORR), were calculated against the measurements. RMSE and MAE are widely used in model evaluations to assess the absolute deviation between the model output and observations, and the combination of these metrics is often required to assess model performance (Chai and Draxler, 2014). BIAS is an indicator to intuitively show the degree of datasets prone to overestimation or underestimation. The CORR can be used to represent the extent to which a dataset satisfy a linear relationship (Su et al., 2018).

## 3 Results and discussion

### 3.1 Scaling behavior

To assess the computational efficiency of the NAQPMS-PDAF described in Section 2.3 in high-performance computing (HPC), we performed a scaling study on the HPC subsystem of the Big Data Cloud Service Infrastructure Platform (BDCSIP) located in Beijing (China). The HPC subsystem of the BDCSIP (HPC-BDCSIP) consists of 313 Intel Xeon scalable gold computation

nodes, and it entirely has 12 520 processors and nearly 88 TB main memory. The cores within one computation node is 40, 157 of which have 384 GB main memory and the others have 192 GB main memory, and runs at 2.50 GHz. All computing nodes are connected by a 100 Gbps EDR (Enhanced Data Rate) InfiniBand network. The data storage subsystem of the BDCSIP can be accessed by all computing nodes through InfiniBand, and the total capacities are more than 25 PB. The HPC-BDCSIP achieves a performance of Linear system package with 1.0 Petaflops.

The scaling of the NAQPMS-PDAF can be investigated through strong and weak scaling tests. A strong scaling is the parallel performance with the overall workload remaining fixed but the number of processors increasing, while the weak scaling is when the workload assigned to each process remains constant with an increase in processors (Sharples et al., 2018). The runtime is expected to decrease with an increasing number of processors in strong scaling, while the runtime is expected to be constant in weak scaling. As the strong scaling properties of the NAQPMS and PDAF have been examined in previous

studies (Wang et al., 2017a; Nerger and Hiller, 2013), the NAQPMS-PDAF, an online coupled system of the NAQPMS and PDAF, is needed to first perform both strong and weak scaling tests to investigate the computational efficiency. To balance





computational resources and representativeness, we performed 24 analysis steps (from 18:00 UTC on 8 April 2019 to 17:00 UTC on 9 April 2019) for the scaling studies. For each compute node, 25 cores were used, which was found to be the best configuration under preliminary tests considering both execution time and memory requirements. Since the decomposition of

the model domain in the MPI is along the longitudinal grids, as discussed in Section 2.3.1, the number of processors of one realization must be set as the value, which can be divided evenly between the number of longitude grids (300 in this study) and the division between the total number of processors and the number of ensemble members. Therefore, in the strong scaling study, we set the ensemble size as a constant, 20, and increased the processors from 100 to 2000 with a total of 10 tests, which are shown in Table 2. The amount of ensembles was risen from 2 to 50 with a sum of 8 tests in the study of weak scaling.

Parallel efficiency (E) (Sharples et al., 2018) can be used to investigate the scalability of the NAQPMS-PDAF:

$$E_{ss}(bq) = \frac{T(b)}{q \cdot T(bq)}, \tag{10}$$

$$E_{ws}(bq) = \frac{T(b)}{T(bq)}, \tag{11}$$

where $E_{ss}(bq)$ is the parallel efficiency representing strong scaling and $E_{ws}(bq)$ is the parallel efficiency representing weak scaling. $b$ is the number of processors used in the base test and $T(b)$ is the execution time with $b$ processors. $bq$ is the number

of processors used in different tests with $bq = n_p$. As shown in Table 2, $b$ is set as 100 in the strong and weak scaling.

Figure 4 shows the execution times per model hour (blue) and strong scaling efficiency (orange) for different processors. The abscissa is the number of processers and corresponding ensemble size, which is set as 20 in the strong scaling test. The entire blue line decreased as an exponential function and a nearly 12-fold decrease in time with a 20-fold increase with processors. When the number of processors was from 100 to 600, the execution time decreased rapidly, and strong scaling

efficiency remained very high (> 0.85). During this period, the speed increased with more processors apportioned to one model realization. When the number of processes was increasing from 1000 to 2000, the execution time showed a slight decrease, and strong scaling efficiency decreased rapidly from ~0.77 to 0.6. During this period, the number of processors for one model realization was further increasing, leading to too much consumption of data transfer in the NAQPMS-PDAF. As a result, the same increase in processors had less improvement in the strong scaling efficiency than that in the former period.

Referring to the results in the strong scaling behavior, 50 processors were used for each model realization in this study. However, the results of the strong scaling behavior shown here were related to the configuration of the NAQPMS-PDAF and the capability of HPC.

Figure 5 and Figure 6 show timing information and weak scaling efficiency for the weak scaling test. The abscissa is the ensemble size and corresponds to the number of processors. The ratio of the number of processors and ensemble members is

a constant, 50, which was acquired on the strong scaling test. The execution time per model hour was averaged during the 24 hr runs described before, which was the same as the strong scaling test. The orange line shows the execution time of model integration, which dominated the overall time. With the ensemble size increasing from 2 to 50, the execution time of



initialization (blue line) and finalization (red line) of the NAQPMS-PDAF increased from 4.6 s to 36.8 s and 4.8 s to 37.9 s, respectively. The execution time of ensemble simulations (orange line) dominated the overall time and increased from 223 s

to 277 s. The execution time of assimilation remained steady and increased from 1.3 s to 1.6 s.

As shown in Fig. 6, the weak scaling efficiency of model integration decreased from 1.0 to 0.8 as the ensemble size increased from 2 to 50. Although the weak scaling efficiency of initialization and finalization decreased rapidly from 1.0 to 0.12, showing slightly poor parallel behavior, the execution time of these two parts only accounted for a small portion of the overall time. The weak scaling behavior of the assimilation step in the NAQPMS-PDAF was the best and decreased from 1.0

to 0.81. In summary, the total weak scaling efficiency decreased from 1.0 to 0.7, which showed comparable results compared with other coupled systems with the PDAF, such as the TerrSysMP-PDAF (Kurtz et al., 2016). As a result, the coupling between the NAQPMS and PDAF worked very well in a technical sense. In the above scaling study, real ensembles were used, and the results may have been affected by load imbalance to some extent due to heterogeneous ensembles. We set the task exclusively when using HPC to avoid the load affected by other users.

**3.2 Evaluation of the data assimilation framework**

**3.2.1 Ensemble performance**

The results of DA analysis and subsequent one-hour simulation depend on the ensemble performance to a large extent. The comparison of the prior RMSE and the prior total spread is utilized to measure the ensemble spread in this study. The prior RMSE is defined as $\sqrt{\frac{1}{P}\sum_{i=1}^{P}\left(y_i^0 - \overline{y}_i^f\right)^2}$, where $y_i^o$ is an assimilated observation, $\overline{y}_i^f$ is the prior ensemble mean of the

observation mapped from the background simulation and $p$ is the number of assimilated observations. The prior total spread is defined as $\sqrt{\frac{1}{P}\sum_{i=1}^{P}\left(\sigma_i^{o2} + \sigma_i^{f2}\right)}$, where $\sigma_i^{o2}$ is the observation error variance and $\sigma_i^{f2}$ is the prior ensemble variance in the observation space. The prior RMSE and the prior total spread should be balanced in a well-calibrated system (Houtekamer et al., 2005).

Figure 7 shows the prior RMSE and the prior total spread of extinction coefficients at 50 m, 150 m, 502 m and 1000 m and

surface PM$_{2.5}$. As we can see in Fig. 7a, these two factors show comparable magnitudes and trends in the surface PM$_{2.5}$ and extinction coefficients at 50 m, indicating that the NAQPMS-PDAF is well balanced. However, the total spreads of extinction coefficients at altitudes of 150 m, 502 m and 1000 m show an insufficient spread. The same scenario occurred on the surface PM$_{2.5}$ in Peng et al. (2017), in which they found that it was affected by heavy pollution with much larger RMSEs of PM$_{2.5}$. The insufficient spread of extinction coefficients at higher altitudes was probably because the initial ensemble was disturbed

mostly near the surface and high altitudes only have a small number of elevated sources. Air pollution at high altitudes is mostly due to transport and secondary formation and seldom has direct sources.



### 3.2.1 Internal check

In this section, an internal check (or called sanity check), a comparison of analyses and subsequent one-hour simulations (or called forecasts) of DA experiments with assimilated observations are performed to characterize the performance of the
NAQPMS-PDAF. The extinction coefficients measured by ground-based lidars and simulated by the NAQPMS-PDAF from the height of the blind zone (300 m) and the height of 4275 m are chosen as sample data.

Figure 8 shows the extinction coefficient scatters from the model versus the ground-based lidar measurements averaged over 5 assimilated sites (DA sites) and 6 verified sites (VE sites) for the four numerical experiments. Figure 9 shows the frequency distribution of the forecasts and the analysis of the corresponding experiments shown in Fig. 8. As shown in Fig. 8a
and Fig. 8i, a variety of scatters are distributed outside the 2:1 line, showing that aerosol extinction coefficients were obviously underestimated in the FR experiment, especially below 1200 m. This indicates that the numerical model has deficiencies in reproducing pollutant plumes, especially inside the boundary layer, compared with lidar measurements. The underestimation of the aerosol vertical distribution, which commonly exists in other CTMs such as POLYPHEMUS and WRF-Chem (Wang et al., 2013; Ma et al., 2020), is attributed to numerous uncertainties in direct sources (e.g., emissions and initial/boundary
conditions), physical and chemical processes concerning the formation of aerosols (e.g., nucleation, condensation, horizontal advection, and vertical diffusion) and model parameterizations, as well as uncertainties in the meteorological field.

As shown in Fig. 8c and Fig. 8g, most scatters showing underestimation in the FR experiment are distributed between the 1:2 line and 2:1 line in the NP-LIDAR experiment. In the NP-LIDAR experiment (Fig. 8c, 8g), the absolute value of the negative BIAS value decreases from 0.20 1/km (FR) to 0.02 (0.04) 1/km, and the CORR rises from 0.33 (FR) to 0.91 (0.72)
in the analysis (forecast). The RMSE value decreases from 0.42 1/km (FR) to 0.16 (0.27) 1/km, and the MAE value decreases from 0.25 1/km (FR) to 0.08 (0.16) 1/km in the analysis (forecast) in the NP-LIDAR experiment. The frequency distribution of extinction coefficients is more squeezed around the value of 0.0 1/km with higher peaks in the NP-LIDAR experiment than in the FR experiment (Fig. 9b, 9e). In the analysis, 46 % (13 %) of the extinction coefficient BIASs are within $\pm 0.1$ 1/km ($\pm$ 0.02 1/km) in the FR experiment, while 78 % (33 %) of the BIAS is achieved within $\pm 0.1$ 1/km ($\pm 0.02$ 1/km) in the NP-
LIDAR experiment, respectively. The 1-hr forecast also shows a positive performance with a relatively poor statistical performance compared with that in the NP-LIDAR experiment due to the attenuation of the impact of DA during the model simulation. This suggests that after assimilating the vertical profiles of extinction coefficients measured by ground-based lidars including vertical profile information, the underestimation is noticeably reduced. Moreover, the performance of NP-LIDAR-PM25 is generally comparable to that of the NP-LIDAR experiment, while the performance of NP-PM25 is also similar to that
of the FR experiment. This indicates that assimilating surface PM$_{2.5}$ measurements has a limited impact on higher layers in the analysis and 1-hr forecast.

Maps of statistical metrics (RMSEs and correlation coefficients) between the modeled and surface-measured PM$_{2.5}$ are presented to investigate the DA performance of the four experiments, which is shown in Figure 10 and Figure 11. The increments of specific metric indicate the difference of this metric between a DA experiment and the FR experiment. The





increments in RMSEs (CORRs) of all sites are negative (positive) in the NP-PM25 experiment, with a range of approximately -60 % to -30 % (30 % to 80 %), and generally, the absolute value of those increments over the southern part of the NCP is larger than those over the northern part of the NCP (Fig. 10d and Fig. 11d). It shows pronounced improvement of directly assimilating surface PM$_{2.5}$ measurements. The increments in RMSEs of most sites are negative in the NP-LIDAR experiment, with a range of approximately -40 % to -5 %, while a few sites have positive increments (Fig. 10e). However, the increments in CORRs of all sites are positive in the NP-LIDAR experiment at more than 20 % (Fig. 11e). These results indicate that only assimilating ground-based lidar measurements can improve the surface PM$_{2.5}$ simulations of most areas by modifying the initial conditions of aerosol vertical profiles, especially under the view of linear trends, but there are still some deficiencies at some sites. The performance of statistical metrics on the surface (RMSEs and CORRs) in the analysis in the NP-LIDAR-PM25 experiment is similar to those in the NP-PM25 experiment because the last observation assimilated in NP-LIDAR-PM25 is the surface PM$_{2.5}$ measurement and the first kind of observation (ground-based lidar measurements) has no direct impact (due to lidar blind zone) on the surface. However, the difference between the NP-PM25 and NP-LIDAR-PM25 experiments occurs in the forecast, which indicates that assimilating lidar measurements has a certain impact on the surface within one model hour and slightly degrades the performance of surface PM$_{2.5}$ simulations (Fig. 10k). It can be concluded that the performance of the NAQPMS-PDAF in simulating surface PM$_{2.5}$ is better than that in the FR experiment after assimilating surface PM$_{2.5}$ measurements, ground-based lidar measurements and both of these measurements. However, the performance of only assimilating surface PM$_{2.5}$ measurements on the surface aerosol simulations is better than that of only assimilating ground-based lidar measurements. This could be explained by the relatively sparser distribution of lidar sites compared with surface PM$_{2.5}$ measurement sites and the uncertainty in the spatial representation of lidar data (Liang et al., 2020), as well as the errors in the lumped variables of extinction coefficients with multiple contributions by different aerosol components. Moreover, the problem can also be attributed to the discordant relationship between aerosol mass concentration and extinction coefficients both in the simulation and measurements, which was noticed by Ma et al. (2020), and a simple bias correction method was proposed to fix this problem. The problem discussed above will be discussed in more detail in a separate study.

### 3.2.2 Independent validation

In this section, the DA performance of the NAQPMS-PDAF on the sites of the surface PM$_{2.5}$ observations and ground-based lidar measurements that are not assimilated is investigated as the independent validation, as well as the independent CALIOP and AERONET measurements.

The extinction coefficients averaged over the 6 VE sites are underestimated with a BIAS of -0.14 1/km, showing a slightly better simulation performance compared with those at the 5 DA sites (Fig. 8i). In the NP-LIDAR experiment, the analysis and subsequent one-hour simulation performance of the four statistical metrics (RMSE of 0.30 1/km and 0.30 1/km, MAE of 0.18 1/km and 0.18 1/km, BIAS of 0.00 1/km and -0.01 1/km, and CORR of 0.55 and 0.53) at VE sites is particularly similar (Fig. 8k, 8o), which shows a significant improvement compared with those in the FR experiment (Fig. 8i). This suggests that the initial chemical conditions at VE sites after the implementation of the ensemble filter of the previous step in the analysis show





good consistency with the lidar observations which are not assimilated. The difference between the analysis and forecast at VE sites can be identified in Fig. 9h and Fig. 9k, where extinction coefficients are more squeezed around a value of 0.0 1/km

with higher peaks in the NP-LIDAR experiment than in the FR experiment. Moreover, the statistical metrics characterizing the absolute deviation (RMSE, MAE and BIAS) of the forecast at DA sites and the analysis at VE sites are similar, and the degree of fitting a linear relationship (CORR) of the forecast at DA sites is better than that in the analysis at VE sites. This indicates that the qualitative attenuation is weaker than the quantitative attenuation after assimilating measurements including aerosol vertical information. In other words, the overall performance of the NAQPMS-PDAF at VE sites in NP-LIDAR is

superior than that in the FR experiment with the spread of DA under the time-varying background error covariance matrix represented by ensembles. At VE sites, the performance of the NP-LIDAR-PM25 experiment is similar to that in the NP-LIDAR experiment and the same as that in the FR and NP-PM25 experiments, which is similar to that at DA sites discussed in Section 3.2.1.

In the NP-PM25 experiment, the spatial distribution of the increases in RMSEs at VE sites is similar to that at DA sites with

an approximately 10% decrease (Fig. 10p). The slight decrease is attributed to the impact of DA attenuation with distance, although the DA sites and VE sites are almost within the same city. In the NP-LIDAR experiment, the increases in RMSEs slightly rise at a few sites (Fig. 10q), and the probable reasons are discussed in Section 3.2.1. The performance of RMSEs and the increments of RMSEs in the NP-LIDAR-PM25 experiment (Fig. 10o and Fig. 10r) are similar to those in NP-PM25, and a related discussion can be seen in Section 3.2.1. The difference between the analysis and forecast at VE sites is comparable

with that at DA sites. It can be concluded that NP-PM25 and NP-LIDAR-PM25 can significantly improve the surface $PM_{2.5}$ simulation, with all sites showing negative (positive) increments in RMSEs (CORRs). However, a few sites show positive increments of RMSEs, while all sites show negative increments in CORRs in the NP-LIDAR experiment. The difference of increment performance of RMSEs between the NP-LIDAR-PM25 (NP-PM25) and NP-LIDAR is larger than that of CORRs. This suggests that the lidar measurements assimilated indeed include authentic vertical distribution information of aerosols,

and the NP-LIDAR experiment can notably improve the aerosol vertical distribution simulations and then improve the surface $PM_{2.5}$ simulations in the numerical model, but the quantification especially on the surface $PM_{2.5}$ mass concentration, needs to be strengthened. Apart from the solutions discussed in Section 3.2.1, a systematic lidar data quality assurance and control scheme (Wang et al., 2020b) is another method to solve this problem and is urgently needed for further research.

Fifty-two CALIPSO orbits are covered within the model domain during the one-month (April 2019) period. However, only

a few CALIOP measurements can be utilized to evaluate the performance of the NAQPMS-PDAF due to sparse coverage and data integrity. Figure 12 shows aerosol extinction coefficient vertical profiles of CALIOP measurements, as well as those in the analysis of the four experiments in regard to 6 orbits with different times. The chosen CALIPSO orbits are mapped in Fig. 12b, and the gray lines denote the part of orbits with missing extinction coefficient data through the vertical profile. The vertical profiles of extinction coefficients of the CALIOP measurements and the analysis of the four experiments at 05:00 UTC 5 April

2019 are shown in Fig. 12a. Although the orbits are slightly covered by the model domain, the only difference between the FR and NP-LIDAR experiments is whether ground-based lidar measurements are assimilated (Fig. 12b). The shape of the analysis





of extinction coefficient profiles from heights of 1300 m to 2200 m in NP-LIDAR is commonly consistent with the independent measurements from the CALIOP, while the background simulation underestimates the aerosol extinction profiles (Fig. 12a). The ground-based lidar assimilation induces a slight overestimation of the vertical profiles of extinction coefficients from the

near surface to a height of 1300 m, whereas it fails to capture the high value at a height of approximately 500 m. The same pattern can be found at 18:00 UTC on 17 April 2019 (Fig. 12d), 05:00 UTC on 26 April 2019 (Fig. 12f) and 18:00 UTC on 27 April 2019 (Fig. 12g). The probable reason is that the aerosol loading is mainly suspended below 1000 m with high uncertainty, especially introduced by spatial aerosol inhomogeneities (Gimmestad et al., 2017). However, except for the underestimation, the FR experiment overestimates the aerosol vertical profiles below approximately 2400 m at 18:00 UTC on 14 April 2019

(Fig. 12c). The overestimation is slightly amended by the NP-LIDAR experiment due to the small cover within the model domain (Fig. 12b). The underestimation of the aerosol vertical profile is found in both the FR and NP-LIDAR experiments at 05:00 UTC on 18 April 2019 (Fig. 12e), with most orbits out of the impact range of DA. In one word, the NP-LIDAR experiment can significantly improve the aerosol vertical profile simulations whether with overestimations or underestimations, especially below approximately 2000 m. Therefore, the evaluation against the CALIOP measurements from NP-LIDAR is

consistent with that from NP-LIDAR-PM25, and the comparison between FR and NP-PM25 experiments also produces the same conclusion.

Apart from space-borne lidar measurements, ground-based AOD measurements are also utilized to independently evaluate the performance of the NAQPMS-PDAF, especially assimilating ground-based lidar measurements. The number of AERONET sites is more than 700 around the world, and AERONET measurements are widely used. While only the AOD

measurements are from the four AERONET sites, the evaluation based on these sites is still a preliminary reference to the performance of the NAQPMS-PDAF. Figure 13 shows the time series of comparison between AERONET AODs and the AODs calculated from the analysis and forecast in the four experiments in this study. The locations of these four sites shown in Fig. 13 are approximately concentrated in Beijing. Referring to the spatial distribution of ground-based lidar shown in Fig. 3, these four AERONET sites are mainly affected by the DA lidar of No. 10 in Langfang, as well as some adjacent surface

PM$_{2.5}$ measurements. Due to the relatively close distance, the DA performance of the NAQPMS-PDAF is similar over these four AERONET sites. Taking the Beijing-PKU site as an example, the AOD measurements show diurnal variability with high AOD at night and relatively low AOD in the daytime (Fig. 13a). The FR and NP-PM25 experiments have good consistency with the AOD measurements when AODs are less than 0.5. However, an AOD underestimation can be identified in the FR and NP-PM25 experiments during the period with relatively high AODs (> 1.0), such as the period from 12:00 LST on 21

April 2019 to 12:00 LST on 24 April 2019. The simulations (analysis and forecast) in the NP-LIDAR and NP-LIADR-PM25 experiments generally agree well with the AERONET AOD measurements, which can capture the relatively high AODs (such as the period from 12:00 LST 21 April 2019 to 12:00 LST 24 April 2019) and low AODs (such as the period from 12:00 LST 6 April 2019 to 12:00 LST 8 April 2019.). In summary, the AOD simulations have a more consistent pattern with the AERONET AOD after assimilating ground-based lidar measurements including aerosol vertical information, while only

assimilating surface measurements has no such advantages.



### 3.3 Vertical profile analysis

Figure 14 shows the vertical distribution of aerosol extinction coefficients averaged over DA sites and VE sites, respectively, in which the vertical profiles of extinction coefficients measured by ground-based lidar (red line) are the output after vertical interpolation. The difference in vertical profiles averaged over DA sites between the FR and NP-PM25 experiments in the
analysis can be found only near the surface with slightly small extinction coefficients in the NP-PM25 experiment (Fig. 14a). This suggests that the assimilation of surface $PM_{2.5}$ measurements can only affect the first layer of the model, as well as the above two layers under the impact of vertical correlation length, and has limited influence on the entire aerosol vertical distribution, which is an inducement to assimilate measurements, including aerosol vertical information, to improve the aerosol vertical profile simulations. The aerosol extinction coefficients decrease with height in the FR and NP-PM25 experiments,
which show large discrepancies compared with the ground-based lidar measurements, especially from 400 m to 1000 m. The discrepancies of aerosol extinction coefficients between the FR (NP-PM25) and NP-LIDAR (NP-LIDAR-PM25) experiments (negative increments) occur at a height of approximately 5000 m, which corresponds to the top height (5235 m) of ground-based lidar measurements assimilated. The extinction coefficients are relatively small at a height of approximately 5000 m in the FR experiment due to a lack of aerosol loading. The negative increments of extinction coefficients between the FR (NP-
PM25) and NP-LIDAR (NP-LIDAR-PM25) experiments, namely, the difference of extinction coefficients, are gradually improved with decreasing height. The most significant improvement occurs at a height of approximately 400 m with extinction coefficients of 0.37 1/km, where the extinction reaches the highest value. The extinction coefficients in the analysis then decrease with decreasing height until a height of 300 m, which is the blind zone height in the NP-LIDAR and NP-LIDAR-PM25 experiments. The discrepancies in aerosol extinction coefficients between the NP-LIDAR and NP-LIDAR-PM25
experiments occur below the lidar blind zone height, in which the extinction coefficients in NP-LIDAR sharply increase and those in the NP-LIDAR-PM25 decrease with the decreasing height. The extinction coefficients near the surface in the NP-LIDAR-PM25 experiment are closer to those in the NP-PM25 experiment than those in the NP-LIDAR experiment. Note that the surface $PM_{2.5}$ simulations in NP-PM25 are superior to those in the NP-LIDAR experiment, which has been evaluated in Sections 3.2.1 and 3.2.2. Therefore, the discrepancies are due to the NP-LIDAR-PM25 experiment assimilating the surface
$PM_{2.5}$ measurement, which makes the aerosol vertical profiles more consistent with the measurements than those in the NP-LIDAR experiment near the surface. The aerosol vertical profile in the analysis in NP-LIDAR-PM25 combines the model output and lidar measurements with the minimum uncertainty (orange line in Fig. 14a), showing aerosol loading at high elevations of approximately 500 m. As shown in Fig. 14b, the difference in extinction coefficients in the one-hour forecast between the FR and NP-PM25 decreases more than that in the analysis (Fig. 14a). The consistency between the forecast and
lidar measurements degrades within one model hour simulation at a height of approximately 700 m. The variability in the forecast is probably due to the attenuation of the impact of DA during the model simulation. The pattern of the vertical profile in the forecast is similar to that in the analysis in the NP-LIDAR experiment. The near-surface part of the aerosol vertical profile in the NP-LIDAR-PM25 experiment is close to that in the NP-LIDAR experiment. This suggests that the temporal





effect of assimilating the surface PM$_{2.5}$ measurements is weaker than that of assimilating the ground-based lidar measurements.

The aerosol vertical profile averaged over VE sites shows a similar shape to that over DA sites, indicating that the measurements at DA sites have good consistency compared with those at VE sites. The aerosol vertical profile in the NP-LIDAR (NP-LIDAR-PM25) experiment shows good consistency with that in the lidar measurement above a height of approximately 500 m, of which the peak extinction value has a discrepancy below that height. The decreasing gradients of extinction coefficients at VE sites from a height of approximately 500 m to the blind zone are steeper than those at DA sties.

This indicates that the peak height becomes low with the attenuation of the impact of DA. Combined with the attenuation of the impact of surface DA, a particular break occurs near the surface in the NP-LIDAR-PM25 experiment (Fig. 14c). The difference in extinction coefficients between the analysis and forecast at VE sites is similar to that at DA sites. Following Su et al. (2020), aerosol vertical structures can be largely classified into three types: well mixed, decreasing with height, and inverse structures. The shape of the vertical structure is totally changed from decreasing with height in the FR experiment to

inverse structures in the NP-LIDAR and NP-LIDAR-PM25 experiments, which may affect the vertical distribution of aerosol radiative forcing (Li et al., 2017b; Su et al., 2020) and is not the focus of this study.

### 3.4 Uncertainties

The heart of DA is dealing with uncertainty (Bannister, 2017). The ensemble members can provide error information in ensemble-based DA. The analysis ensemble spread, estimated as the standard deviation of the simulations across the ensemble,

can be used as an indicator of the analysis uncertainty on the estimated observations mapped by the state vectors in the ensemble-based DA framework (Jr et al., 2007; Miyazaki et al., 2012). The analysis uncertainty is concerned with errors in input data such as emissions, the representation of physical and chemical processes in numerical models, assimilated measurements and so on (Miyazaki et al., 2020). The ensemble spread of extinction coefficients simulated across the 20 ensemble members in the FR and NP-LIDAR-PM25 experiments is shown in Figure 15, and the height from the surface to

1700 m is divided into three bins (50 m, 64 m to 502 m, and 550 m to 1700 m) to present different characteristics of uncertainty. The color scale bar is set to the same height bin and is different at the different height bins. The analysis spread in FR at 50 m (considered the surface in the model) is mainly characterized by a large spread in a few point source regions due to perturbations in emissions (Fig. 15b). The direct effect (assimilating surface measurement) and indirect effect (assimilating ground-based lidar measurement) of posterior fields after DA reduce the uncertainty represented by the analysis spread in the

NP-LIDAR-PM25 experiment, which is vividly shown in Fig. 15c. The ensemble spread in the FR experiment shows a combination of emission-related and transport-related uncertainties from 64 m to 502 m (Fig. 15d). We see changes in the structure of the ensemble spread in NP-LIDAR-PM25, with clear indications of a reduction in uncertainties (Fig. 15e). The analysis spread with uniform spatial distribution averaged from heights of 550 m to 1700 m is less affected by surface emissions and more affected by transport-related uncertainties, such as horizontal advection in the FR experiment (Fig. 15f). The average

analysis ensemble spread from 550 m to 1700 m over the middle and southern NCP, which is the coverage area of lidar DA, is significantly reduced in the NP-LIDAR-PM25 experiment (Fig. 15g). The average profiles of the analysis spread are



calculated from the surface to a height of 3570 m in the FR and NP-LIDAR-PM25 experiments, which are shown in Fig. 15a. The ensemble spread decreases with height in the FR experiment. The reduction of the ensemble spread occurs at a height of approximately 2500 m and increases with decreasing height. It demonstrates that the analysis of ensemble filter after

assimilating ground-based lidar measurements converges to a true state.

## 4    Conclusions and outlook

In this paper, we couple the atmospheric chemistry-transport model NAQPMS with the PDAF online for the first time to establish a high-performance ensemble filter system NAQPMS-PDAF to mainly investigate the impact of assimilating measurements, including aerosol vertical information. We examine the computational efficiency and DA performance of the

NAQPMS-PDAF on the analysis and subsequent one-hour simulations after assimilating one-month-long aerosol extinction coefficient profiles measured by 5 ground-based lidars (6 ground-based lidar measurements are used as evaluation) during April 2019 and the concurrent hourly surface $PM_{2.5}$ measurements over the NCP with four experiments (FR, NP-PM25, NP-LIDAR and NP-LIDAR-PM25). Except for the lidar and surface $PM_{2.5}$ measurements, which are not assimilated, the AERONET AODs and the vertical profiles of extinction coefficients measured by the CALIOP are utilized as independent

validations. The characteristics of aerosol vertical profiles and the uncertainties in the NAQPMS-PDAF are also probed in detail.

The coupling between the NAQPMS and PDAF is implemented in a fully integrated fashion with data exchange performed via main memory, which avoids frequently reading and writing model restart files. Two levels of parallelization are introduced to perform ensemble simulations running fully parallel and to implement the filter algorithm efficiently. The arrangement of

the dimensional order of state vectors during the ensemble filter is especially designed to allow the submatrix to cut along the longitudinal direction with each slice containing full variable information.

The scaling tests on the massively parallel HPC BDCSIP are performed first, where the strong scaling shows that 50 processors per model realization is the optimal configuration synthetically considering strong scaling efficiency and the increment in computational load, and the weak scaling reveals that the NAQPMS-PDAF runs efficiently with the number of

processors from 100 to 2500 with the weak scaling efficiency only decreasing from 1.0 to 0.7. This scaling study indicates that online coupling works very well in a technical sense. The ensemble performance is then evaluated on which all the following results depend. The NAQPMS-PDAF is well balanced near the surface and shows a slightly insufficient spread at high altitude due to the disturbed emissions mainly near the surface, which is evaluated by the ensemble members.

The numerical model without DA has deficiencies in reproducing pollutant plumes, especially inside the boundary layer,

which shows an obvious underestimation in the FR experiment that commonly exists in other CTMs. Compared with the assimilated ground-based lidar measurements as internal checks, the underestimation of the aerosol extinction coefficients is remarkably improved in the analysis of NP-LIDAR and NP-LIDAR-PM25 experiments with the BIAS changing from -0.20 1/km to -0.02 1/km, and the correlation coefficients increasing from 0.33 to 0.91 averaged at sites with DA. The statistical



performance in the subsequent one-hour simulation is always slightly weaker than that in the analysis due to the attenuation

of the impact of DA. Only assimilating surface measurements can directly improve the surface $PM_{2.5}$ simulations but has limited influence on high elevations. However, only assimilating ground-based lidar measurements can mainly improve the surface $PM_{2.5}$ simulations with a slightly weaker performance than that in assimilating surface $PM_{2.5}$ measurements. This could be explained by the relatively sparser distribution of lidar sites compared with surface $PM_{2.5}$ measurements and the uncertainties in the spatial representation of lidar data, as well as the errors in the lumped variables of extinction coefficients

with multiple contributions by different aerosol components. Moreover, the problem can also be attributed to the discordant relationship between aerosol mass concentration and extinction coefficients both in the simulation and measurements. In one word, the performance of the NAQPMS-PDAF to simulate surface $PM_{2.5}$ in assimilating surface $PM_{2.5}$ measurements, ground-based lidar measurements and both of these measurements is better than that in the FR experiment. These results indicate that the NAQPMS-PDAF system operates successfully.

In the independent validation of lidar measurements, the qualitative attenuation is weaker than the quantitative attenuation after assimilating measurements, including aerosol vertical information, when comparing the performance on the forecast at DA sites with that on the analysis at VE sites. The lidar measurements assimilated indeed include authentic vertical distribution information of aerosols, and the NP-LIDAR experiment can notably improve the aerosol vertical distribution simulations and then improve the surface $PM_{2.5}$ simulations in the numerical model, but the quantification, especially on the surface $PM_{2.5}$ mass

concentration, needs to be strengthened. A systematic data quality control of lidar measurements is urgently needed to solve this problem in future research. The aerosol extinction coefficients measured by the CALIOP, which has sparse coverage and limited data integrity, are utilized as independent evaluations, and it is found that assimilating ground-based lidar can significantly improve both the overestimation and underestimation of the extinction values, especially below approximately 2000 m. AODs measured by the four AERONET sites, which are approximately concentrated in Beijing, are also used to

independently validate the performance. Lidar DA can have a more consistent pattern with the AERONET measurements, while only assimilating surface measurements has no such advantages.

In the vertical profile analysis, the aerosol extinction coefficients decrease with height in the FR and NP-PM25 experiments, which presents large discrepancies compared with the ground-based lidar measurements, especially from 400 m to 1000 m. The negative increments of extinction coefficients between the FR (NP-PM25) and NP-LIDAR (NP-LIDAR-PM25)

experiments gradually improve with decreasing height. The most significant improvement occurs at a height of approximately 400 m with extinction coefficients of 0.37 1/km, where the extinction reaches the highest value. Although assimilating surface $PM_{2.5}$ measurements has a limited impact on the aerosol vertical profile, the correction in the NP-LIDAR-PM25 experiment occurs near the surface and makes the aerosol vertical profiles more consistent with the measurements than those in the NP-LIDAR experiment. Assimilating ground-based lidar measurements can have a larger temporal impact than assimilating

surface measurements. In addition, the height of the peak extinction coefficient value decreases within the one-hour forecast due to the attenuation of the impact of DA.



Finally, the uncertainties of the NAQPMS-PDAF are examined. The direct impact (assimilating surface $PM_{2.5}$ measurement) and indirect impact (assimilating ground-based lidar measurements) of posterior fields after DA reduce the uncertainty represented by the analysis spread in the NP-LIDAR-PM25 experiment, which is the emission-related ensemble spread at 50

m, and a combination of emission-related and transport-related ensemble spreads averages from 64 m to 502 m, and the transport-related ensemble spread averages from 550 m to 1700 m. It demonstrates that the analysis of ensemble filter after assimilating ground-based lidar measurements converges to a true state.

The proposed NAQPMS-PDAF can significantly improve the aerosol vertical profile simulations and has a large potential to allow further study of the impact of aerosol vertical distribution. In future work, we plan to investigate the key factors mainly

impacting the surface $PM_{2.5}$ simulations after assimilating measurements, including vertical information, as the performance of lidar assimilation in this study shows a limited positive influence on the surface $PM_{2.5}$ simulations. We also plan to further expand the state vectors, and the measurements assimilated as the NAQPMS-PDAF is a modularized DA system with good extendibility. This will allow us to jointly assimilate surface, ground-based and space-borne measurements for a better three-dimensional characterization of aerosol components, as well as gaseous pollutants such as $SO_2$, $NO_x$ and CO.

**Code and data availability**

The source codes of NAQPMS-PDAF are available online via ZENODO (https://doi.org/10.5281/zenodo.5650639). Please contact Haibo Wang (wanghaibo@mail.iap.ac.cn) to obtain the observation data used in NAQPMS-PDAF.

**Acknowledgements**

We would like to acknowledge the support from the Strategic Priority Research Program of the Chinese Academy of Sciences

(Grant No. XDA19040203), the National High Technology Research and Development Program of China (No. 2019YFC214802), and the Young Talent Project of the Center for Excellence in Regional Atmospheric Environment, CAS (CERAE201803). The author Ting Yang gratefully acknowledges the Program of the Youth Innovation Promotion Association (CAS). We thank the Big Data Cloud Service Infrastructure Platform (BDCSIP) for providing compute resources. This research is supported by the National Key Scientific and Technological Infrastructure project "Earth System Science

Numerical Simulator Facility". We thank the anonymous reviewers for their constructive suggestions that helped improve the manuscript.

**Author contributions**

Haibo Wang implemented PDAF with NAQPMS, performed numerical experiments and carried out the analysis. Ting Yang and Zifa Wang provided scientific guidance for the article structure design. Ting Yang, Zifa Wang and Xueshun Chen provided





valuable suggestions for this article. Jianjun Li, Wenxuan Chai and Guigang Tang provided lidar and surface PM$_{2.5}$ data. Lei Kong provided the code of perturbing the initial emissions. Xueshun Chen provided help for the model code. Haibo Wang wrote the paper and all listed authors have read and approved the final manuscript.

**Competing interests**

The authors declare that they have no conflicts of interest.

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





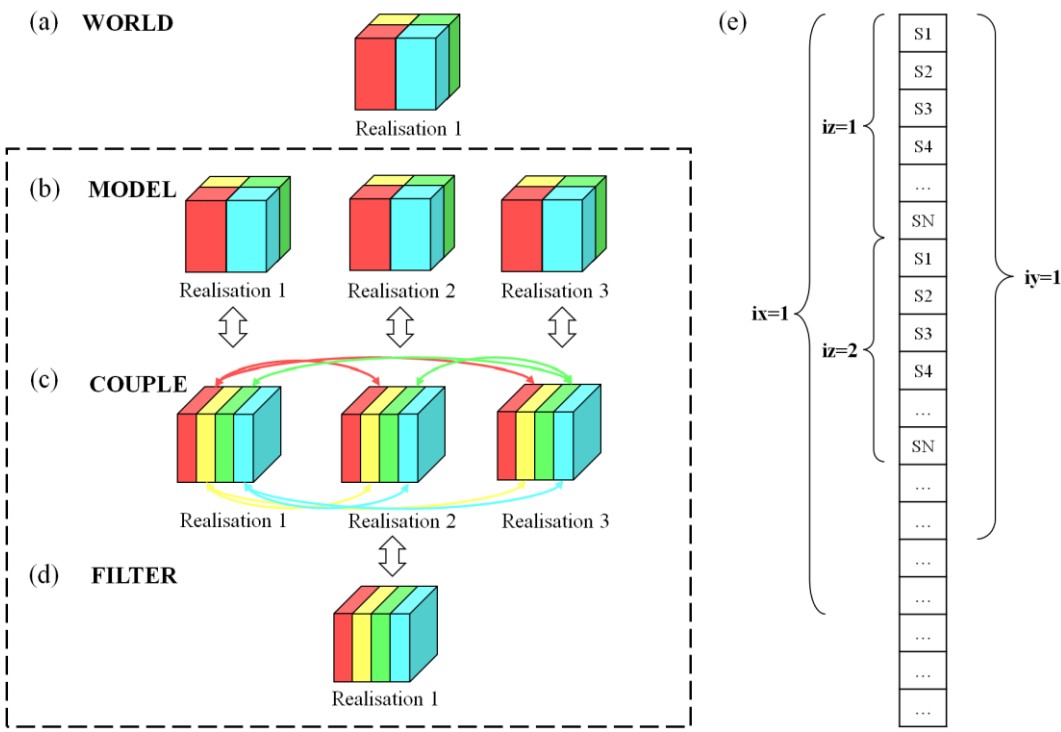

**Figure 1.** Example configuration of MPI communicators for one model realization in the NAQPMS (a) and three model realizations in the NAQPMS-PDAF of the communicator MPI_COMM_MODEL (b), MPI_COMM_COUPLE (c) and MPI_COMM_FILTER (d). The four colors represent the decomposition of processors for different aims. The area outlined by black dashed lines is the NAQPMS-PDAF. The dimension of the state vectors is also shown (e).

| WORLD | | FILTER | | MODEL | | COUPLE | |
|:---:|:---:|:---:|:---:|:---:|:---:|:---:|:---:|
| task | rank | task | rank | task | rank | task | rank |
| 1 | 0 | 1 | 0 | 1 | 0 | 1 | 0 |
| 1 | 1 | 1 | 1 | 1 | 1 | 2 | 0 |
| 1 | 2 | 1 | 2 | 1 | 2 | 3 | 0 |
| 1 | 3 | 1 | 3 | 1 | 3 | 4 | 0 |
| 1 | 4 | | | 2 | 0 | 1 | 1 |
| 1 | 5 | | | 2 | 1 | 2 | 1 |
| 1 | 6 | | | 2 | 2 | 3 | 1 |
| 1 | 7 | | | 2 | 3 | 4 | 1 |
| 1 | 8 | | | 3 | 0 | 1 | 2 |
| 1 | 9 | | | 3 | 1 | 2 | 2 |
| 1 | 10 | | | 3 | 2 | 3 | 2 |
| 1 | 11 | | | 3 | 3 | 4 | 2 |

**Figure 2.** Example of the processor layout of the NAQPMS-PDAF.





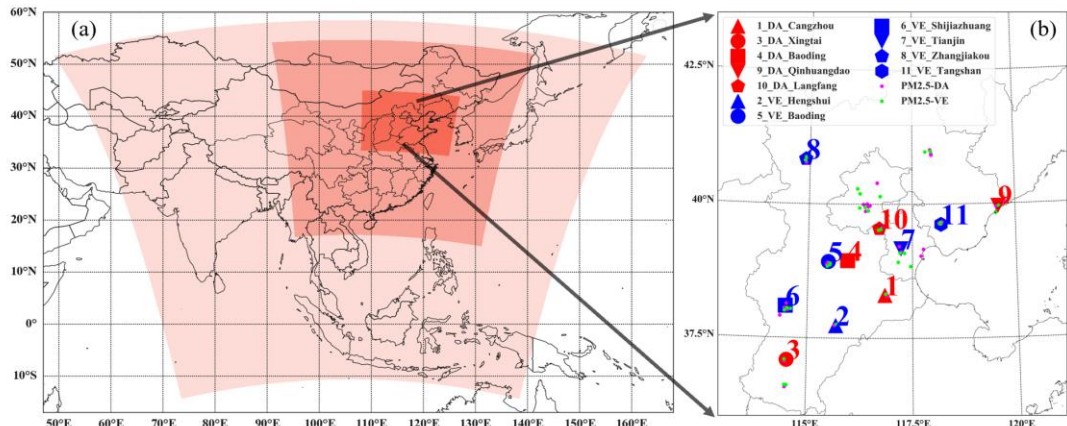

**Figure 3.** The model domain in the WRF simulation (a) and the location of observation stations (b). Three red regions with
different transparencies identify three mode domains (a). Red dots denote ground-based lidar sites that are assimilated, while
blue dots denote verified sites (b). Similarly, pink dots denote assimilated surface PM$_{2.5}$ sites, while green dots denote verified
sites. The serial number denotes ground-based lidar sites.

**Table 1.** Summary of the Experimental design in this study.

| Experiments | PM$_{2.5}$ DA | Ground-based lidar DA |
|:---:|:---:|:---:|
| FR | No | No |
| NP-PM25 | Yes | No |
| NP-LIDAR | No | Yes |
| NP-LIDAR-PM25 | Yes | Yes |



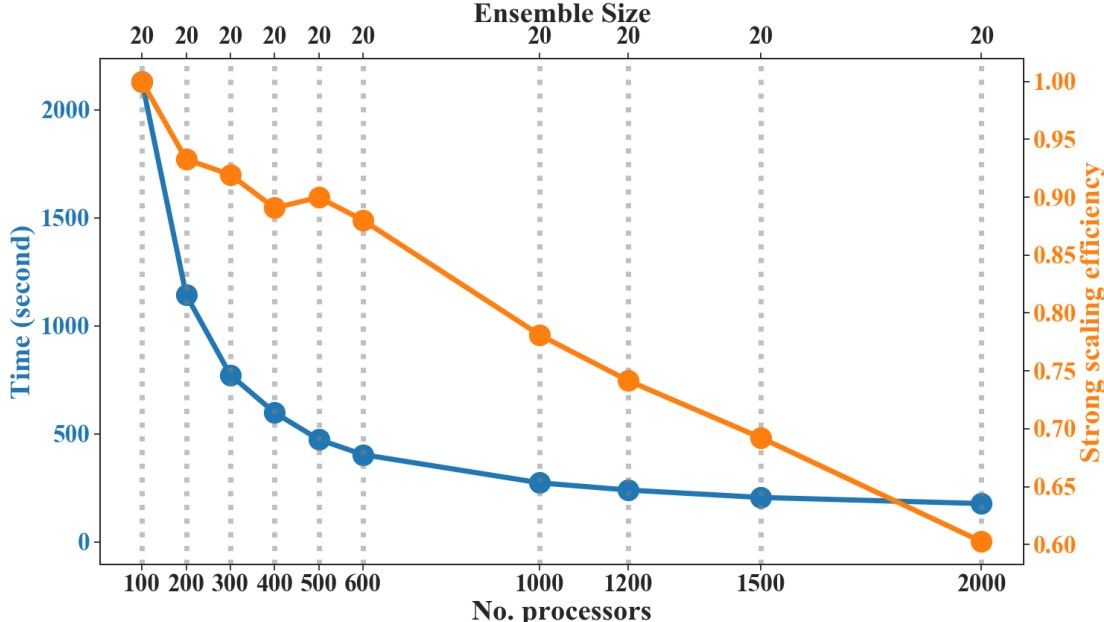

**Figure 4.** Timing information (blue line) and scaling behavior (orange line) for the NAQPMS-PDAF for a strong scaling test. The ensemble size is a constant of 20 and increases the processors from 100 to 2000 with a total of 10 tests.





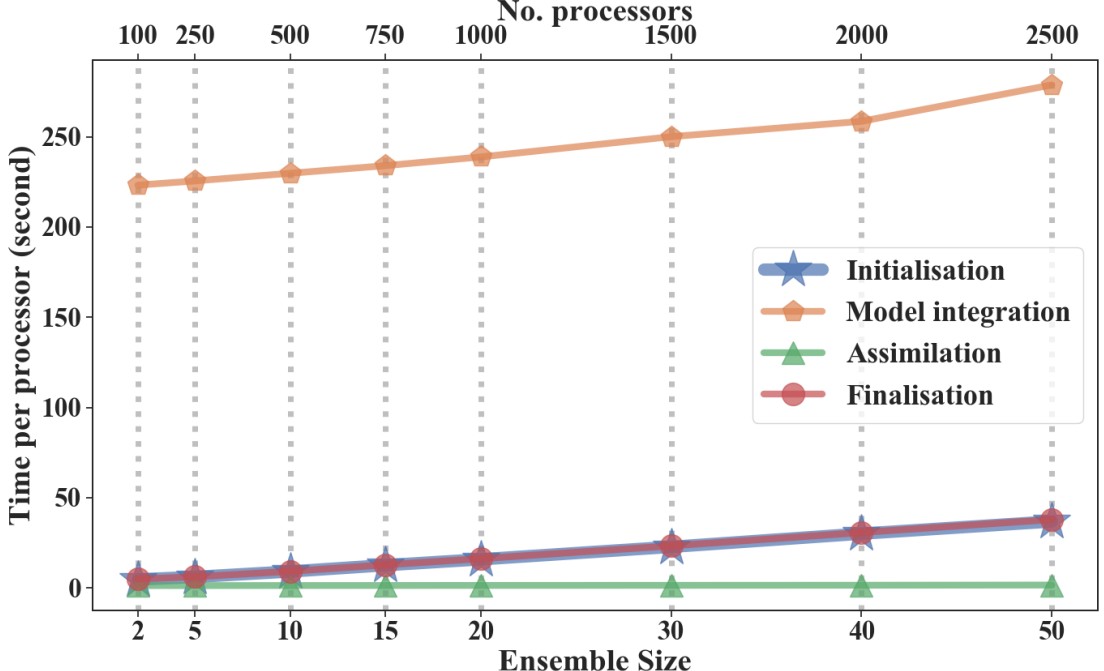

**Figure 5.** Timing information for the NAQPMS-PDAF for a weak scaling test. The blue line shows the execution time for initialization. The orange line shows the execution time for model integration. The green line shows the execution time for assimilation. The red line shows the execution time for finalization. 50 processors are apportioned to each ensemble member.

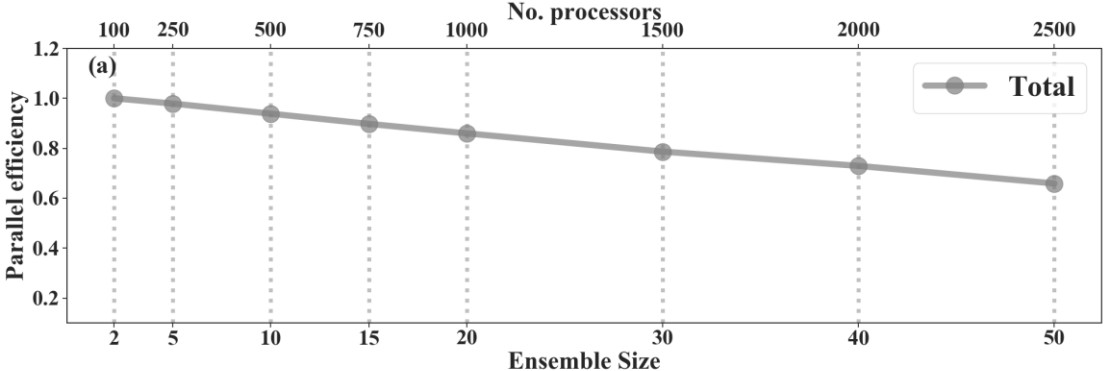

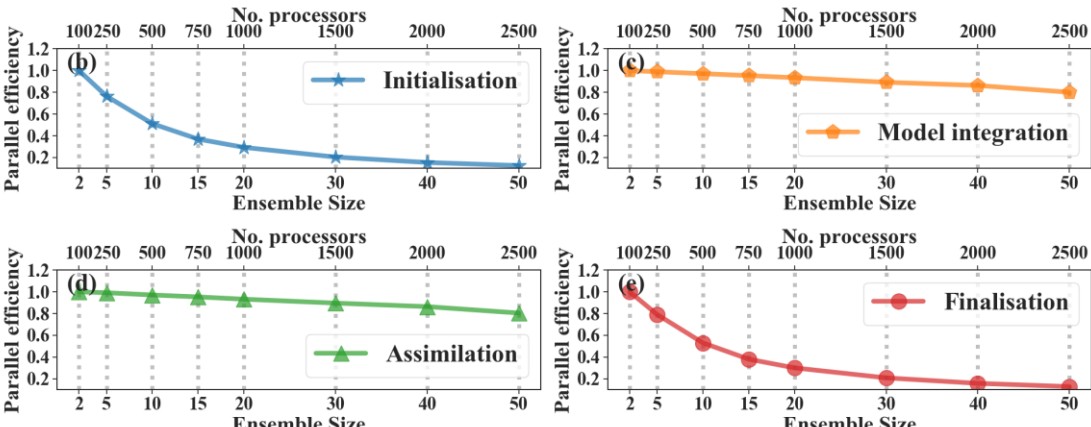

**Figure 6.** Weak scaling behavior for the NAQPMS-PDAF. The gray line shows the parallel efficiency for all processes. The blue line shows the parallel efficiency for initialization. The orange line shows the parallel efficiency for model integration. The green line shows the parallel efficiency for assimilation. The red line shows the parallel efficiency for finalization. 50 processors are apportioned to each ensemble member.




**Table 2.** Number of realizations, compute nodes and processors utilized for the study of strong and weak parallel efficiency for the NAQPMS-PDAF. The number of cores within one compute node is set as 25.

| | No. realizations | No. computer nodes | No. processors |
|---|---|---|---|
| Strong scaling | 20 | 4 | 100 |
| | 20 | 8 | 200 |
| | 20 | 12 | 300 |
| | 20 | 16 | 400 |
| | 20 | 20 | 500 |
| | 20 | 24 | 600 |
| | 20 | 40 | 1000 |
| | 20 | 48 | 1200 |
| | 20 | 60 | 1500 |
| | 20 | 80 | 2000 |
| Weak scaling | 2 | 4 | 100 |
| | 5 | 10 | 250 |
| | 10 | 20 | 500 |
| | 15 | 30 | 750 |
| | 20 | 40 | 1 000 |
| | 30 | 60 | 1 500 |
| | 40 | 80 | 2 000 |
| | 50 | 100 | 2 500 |

1195

1200

1205



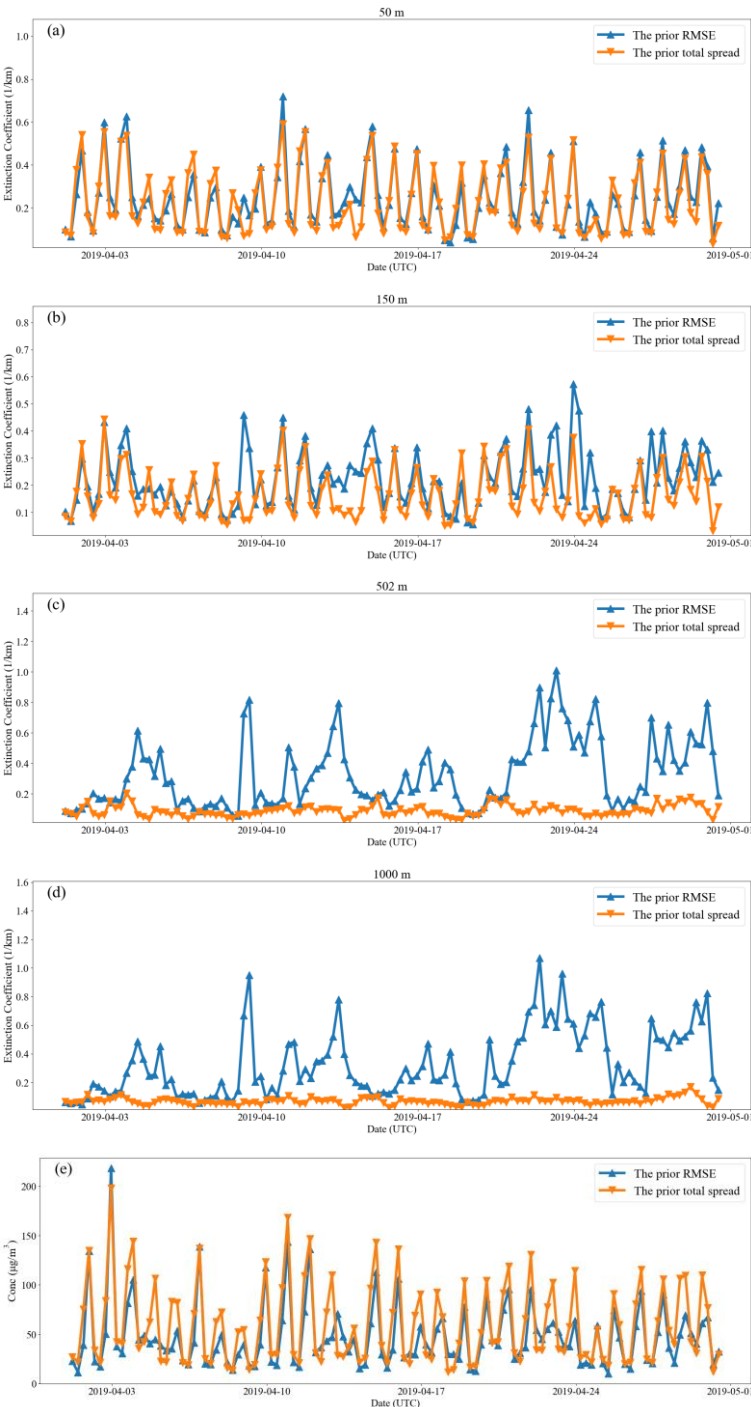

1210

**Figure 7.** Time series of prior RMSE and total spread over all observations for (a) extinction coefficients at 50 m, (b) extinction coefficients at 150 m, (c) extinction coefficients at 502 m, and (d) the surface PM$_{2.5}$



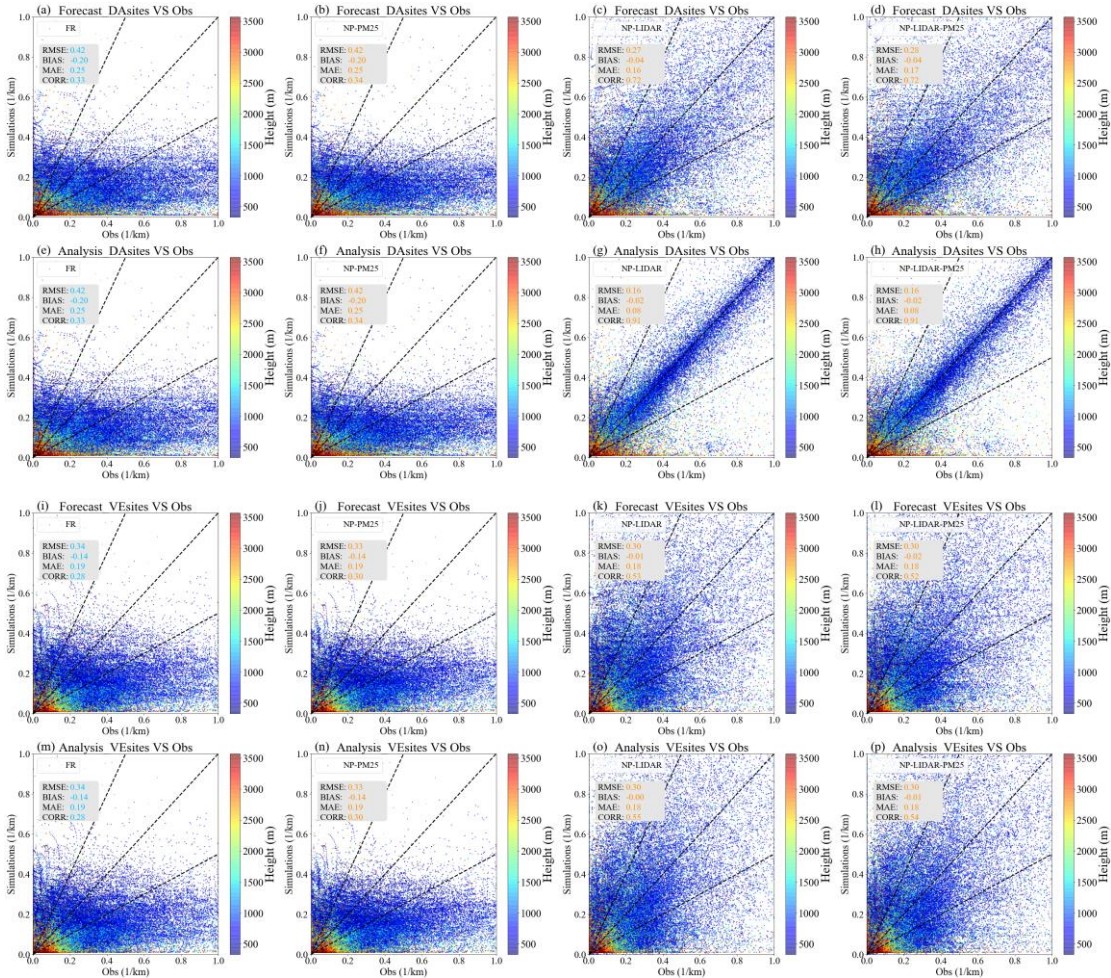

**Figure 8.** Scatter plots of the modeled hourly extinction coefficients at 550 nm versus the ground lidar hourly aerosol extinction coefficients at 532 nm (1/km) of forecasts of FR (a)/(i), forecasts of NP-PM25 (b)/(j), forecasts of NP-LIDAR (c)/(k), forecasts of NP-LIDAR-PM25 (d)/(l), analysis of FR (e)/(m), analysis of NP-PM25 (f)/(n), analysis of NP-LIDAR (g)/(o) and analysis of NP-LIDAR-PM25 (h)/(p), which are averaged among DA sites/VE sites. The three dashed black lines correspond to the 1:2, 1:1 and 2:1 lines in each panel.





**Figure 9.** Frequency distributions of BIAS of forecasts of NP-PM25 versus FR (a)/(g), forecasts of NP-LIDAR versus FR (b)/(h), forecasts of NP-LIDAR-PM25 versus FR (c)/(i), analysis of NP-PM25 versus FR (d)/(j), analysis of NP-LIDAR versus FR (e)/(k) and analysis of NP-LIDAR-PM25 versus FR (f)/(l), which are averaged among DA sites/VE sites.



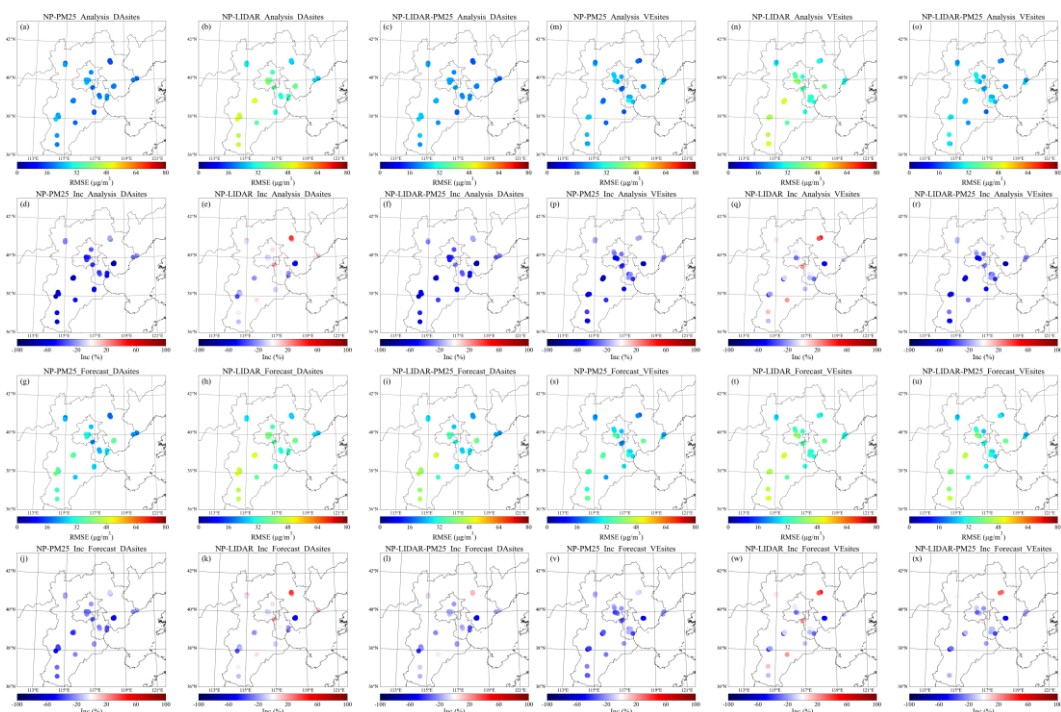

**Figure 10.** Spatial distributions of the RMSEs (1/km) of the surface PM$_{2.5}$ analysis (a, b, c), the increments in the RMSEs of the surface PM$_{2.5}$ analysis (d, e, f), the increments in the RMSEs of the surface PM$_{2.5}$ forecast (g, h, i) and the increments in the RMSEs of the surface PM$_{2.5}$ forecast (j, k, i) for the NP-PM25, NP-LIDAR, and NP-LIDAR-PM25 experiments among DA sites. Fig. (m)-(x) is the same as Fig. (a)-(l) but among VE sites.





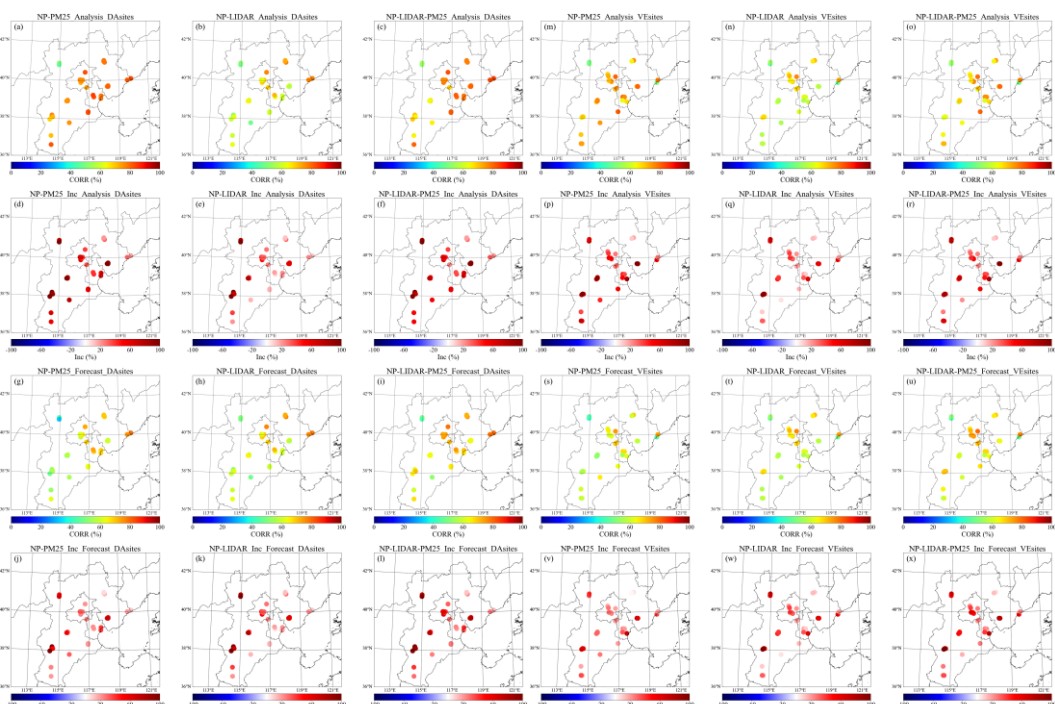

**Figure 11.** Same as Figure 10 but with CORR.

1245



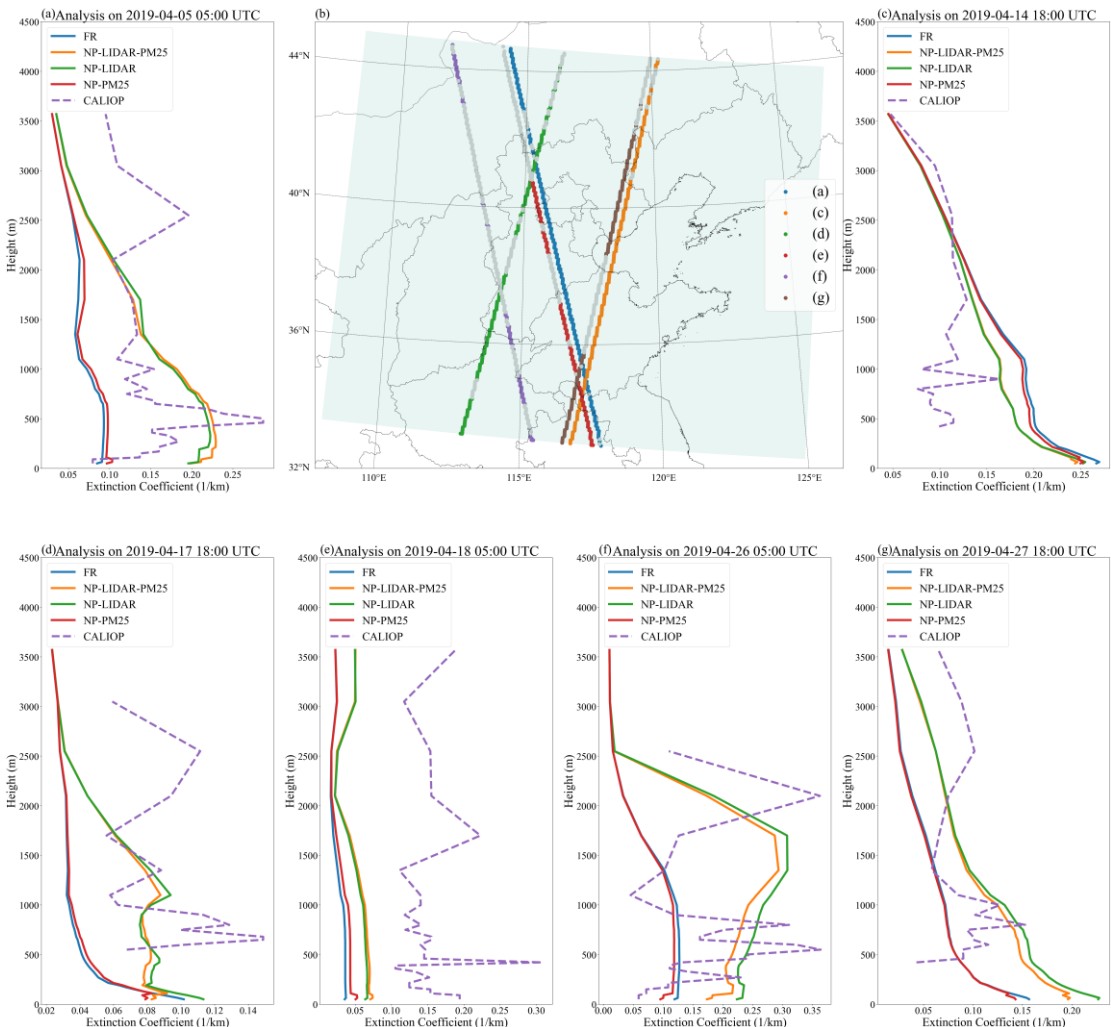

**Figure 12.** The extinction coefficient vertical profile of CALIOP measurement and analysis of the experiment of FR, NP-PM25, NP-LIDAR, and NP-LIDAR-PM25 at 05:00 UTC 5 April 2019 (a), 18:00 UTC 14 April 2019 (c), 18:00 UTC 17 April 2019 (d), 05:00 UTC 18 April 2019, 05:00 UTC 26 April 2019 (f) and 18:00 UTC 27 April 2019 (g). The CALIPSO orbit paths are also shown (b). The shaded area denotes the model domain, and the gray lines (b) denote the part of orbits with missing extinction coefficient data through the vertical profile.



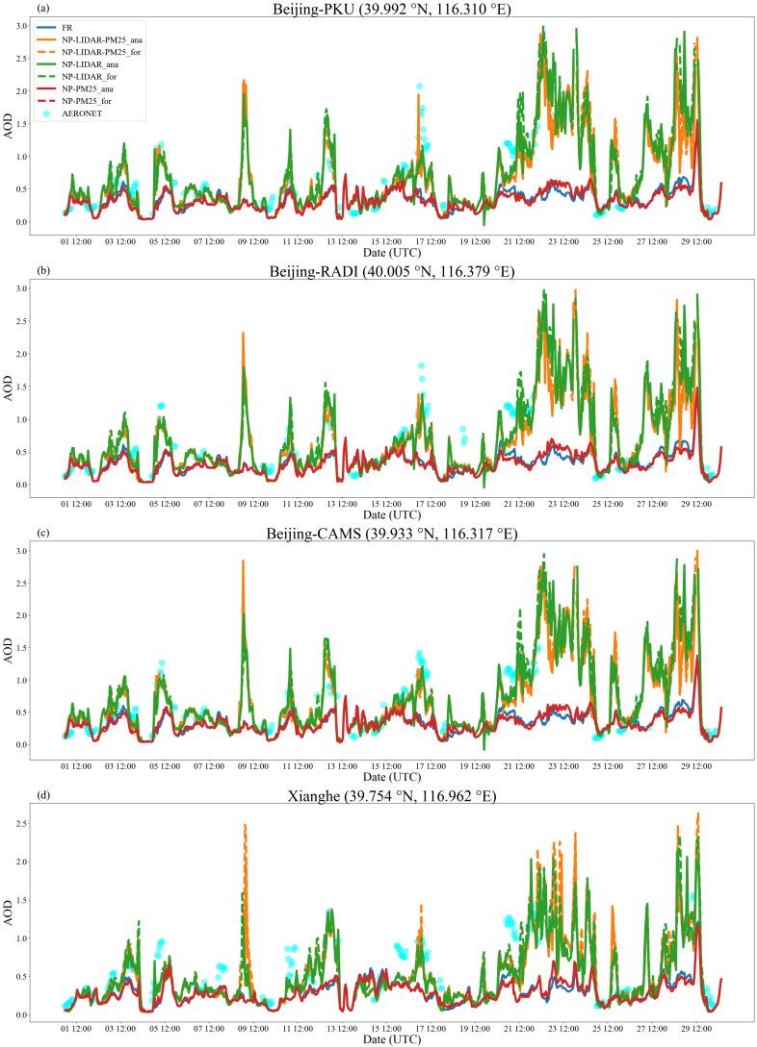

**Figure 13.** Hourly time series of AOD measured by AERONET and in the analysis and forecast of the experiment of FR, NP-PM25, NP-LIDAR and NP-LIDAR-PM25 at the AERONET site of Beijing-PKU (a), Beijing-RADI (b), Beijing-CAMS (c) and Xianghe (d).

1255



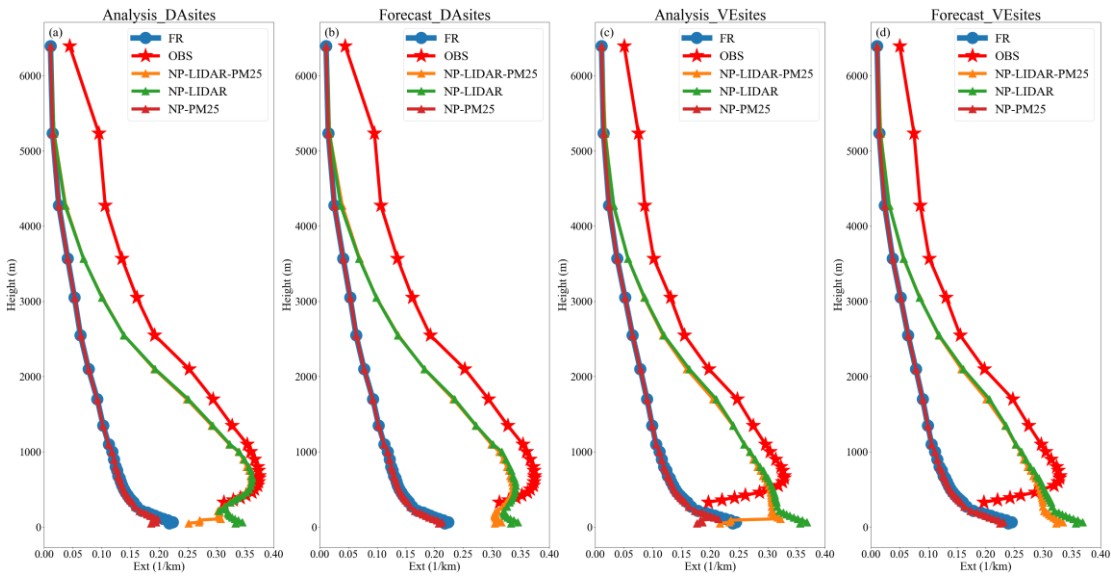

**Figure 14.** Vertical distribution of aerosol extinction coefficients in the analysis averaged among DA sites (a), forecast averaged among DA sites (b), analysis averaged among VE sites (c) and forecast averaged among VE sites (d) in the FR, NP-PM25, NP-LIDAR and NP-LIDAR-PM25 experiments and ground-based lidar measurements.





**Figure 15.** Vertical profile of the analysis ensemble spread of extinction coefficients averaged among DA sites in the FR and NP-LIDAR-PM25 experiments (a). The spatial distribution of the analysis ensemble spread of extinction coefficients at 50 m in the FR experiment (b), at 50 m in NP-LIDAR-PM25 (c), averaged from 64 m to 502 m in FR (d), averaged from 64 m to 502 m in NP-LIDAR-PM25 (e), averaged from 550 m to 1700 m in FR (f) and averaged from 550 m to 1700 m in NP-LIDAR-PM25. All results are averaged over 1-30 April 2021.