# Peer review of "An aerosol vertical data assimilation system (NAQPMS-PDAF v1.0): development and application"

_Geoscientific Model Development, 2021_

## Author Comment (AC1)

The authors appreciate the reviewers for the effort to review our manuscript and to provide constructive comments. As suggested, we carefully revised the manuscript thoroughly according to the valuable advices. Listed below are our point-by-point responses in blue to the reviewer's comments in black. The modifications corresponding to the comments and the revised language and grammars in the manuscript are marked in red.

**Anonymous Referee #1**

**General comments:** This paper describes a chemistry-transport model that is implemented with a data assimilation compartment using the Parallel Data Assimilation Framework. First of all, I would admit that I don't expertise in the atmosphere chemistry area, all my comments are from the data assimilation with PDAF. As for as I know, the online data assimilation approach with PDAF is first thoroughly described in Nerger et al., (2019) GMD paper, where the structure, the algorithm, the implementation are shown based on a climate model AWI-CM. Actually, such implementation has been widely used also for other ocean models such as FESOM, MITgcm etc. Another group like C-Coupler (Liu et al., 2020 GMD) provides similar data assimilation functions as well. I found this research is comprehensive with both technical and experimental perspectives. The results are clearly present and well organized. I would recommend acceptance after minor revisions. In general, I think the paper is well written with adequate evidence for their results.

**Reply:** We thank the reviewer for the positive assessment and constructive suggestions of our manuscript.

**Comment 1:** As for the paper structure, I would suggest the authors rephrase 2.3.1 and 2.3.2, i.e., the technical parts. Most of these implementation details are already well described in Nerger et al., 2019. I didn't see too many differences compared to Nerger's. Currently, it's rather a repetition. Please cite this paper directly and show your differences to condense the context. For 2.3.2, again, not necessarily a repetition of these algorithm details, which are well-known in amount of studies. The authors should concentrate on things that are distinguishable from others' work. For example,

localization radius, whether it is sensitive to your configuration (I see the authors cite two other researches, but actually these are different stories if the configuration changes based on my practices), are you using same localization radius for different observations; forgetting factor, which value is set, why is that; things like that.

**Reply:**

Firstly, we agree with that our manuscript should focus on the differences compared to other studies. The first part of Sec. 2.3.1 is the implementation of two level of parallelization based on MPI. We have cut out redundant descriptions but not deleted the content altogether because it is the first time to introduce PDAF into the study area of CTM. The second part of Sec. 2.3.1 is the main program flow. The third part of Sec. 2.3.1 is the description of the dimension of the state vectors used in PDAF which is specially redesigned in our work. The fourth part of Sec. 2.3.1 is two modules dealing with the data transfer before and after the time loop of the NAQPMS-PDAF which are also designed in our work. In Sec. 2.3.2, we have deleted the description of forecast step and cited the corresponding literature. The analysis step of ESTKF is retained because ESTKF is recently developed especially used in the area of CTM and the difference between ESTKF and other deterministic ensemble filter is mainly the computation of the ensemble transformation in the error subspace. Another anonymous reviewer wants more information about the PDAF and the ESTKF algorithm, so the revision above is the result of both considering two reviewers' comments. Please refer to the revised manuscript Line 174-282.

Secondly, two kinds of observations (surface PM2.5 mass concentration and vertical profiles of aerosol extinction coefficients measured by ground-based lidar) are assimilated into NAQPMS-PDAF in this study. We set the localization radius as 200 km for both observations. For surface PM2.5 concentration, we follow Kong et al. (2020) and set the localization radius as 200 km, because the kind of observation, the atmospheric chemistry-transport model (NAQPMS) as well as the ensemble filter algorithm of this study is same as their work. Therefore, we here focus the localization radius of ground-based lidar and forgetting factor of the data assimilation system, which are set as 200 km and 0.96 in this study. The setting of forgetting factor is omitted in

the original manuscript and have been revised. Please refer to the revised manuscript Line 279-282.

The several sensitivity tests have been made to supplement the setting of these two data assimilation parameters in this study. We refer to Gillet-Chaulet (2020) about the period chosen for sensitivity tests when assimilating real observation under limited computational resources. For sensitivity tests, we choose the study period from 00:00 UTC 23 April to 05:00 UTC 23 April 2019 with abundant pollution plume measured by the ground-based lidar. A series of sensitivity tests are performed with the localization radius (5 km, 50 km, 100 km, 150 km, 200 km, 250 km, 300 km and 400 km) and forgetting factor (0.6, 0.7, 0.8, 0.9, 0.92, 0.94, 0.96, 0.98 and 1.0). The configuration of data assimilation is same as the NP-LIDAR-PM25 experiment in the manuscript expect for the study period. The results of sensitivity tests are evaluated by the VE sites (the ground-based lidar measurements not assimilated) of model domain. Following Nerger (2015) and Nerger (2021), Figure S1 and Figure S2 show the time-mean RMSE and Pearson correlation coefficient for the sensitivity tests, respectively. As we can see in Figure S1, RMSE of aerosol extinction coefficients converges for all combinations of localization radius and forgetting factor. The minimum RMSE of 0.36 1/km is obtained for localization radius of 200 km and forgetting factor of 0.96, 0.98 and 1.0. The maximum Pearson correlation coefficient of 0.75 is obtained for localization radius of 150 km and 200 km and is not sensitive to forgetting factor when forgetting factor is larger than 0.9. Forgetting factor is used to inflate the forecast covariance matrix to reduce under-sampling issues, especially in the long run (Pham et al., 1998). Although the statistical results vary slightly with forgetting factor due to relatively short run time, the combined results of RMSE and Pearson correlation coefficient can provide the optimal parameters in the series of sensitivity tests. To sum up, the forgetting factor of 0.96 and localization radius of ground-based lidar of 200 km is the most optimal parameters. Moreover, the setting of localization radius is same as Cheng et al. (2019) performing ensemble filter to assimilating lidar measurements, and is also close to Ma et al. (2020) performing ensemble filter to assimilating aerosol extinction coefficient profiles measured by ground-based lidar.

[Figure]

**Figure S1.** RMSE for sensitivity tests with localization radius of ground-based lidar (5 km, 50 km, 100 km, 150 km, 200 km, 250 km, 300 km and 400 km) and forgetting factor of NAQPMS-PDAF (0.6, 0.7, 0.8, 0.9, 0.92, 0.94, 0.96, 0.98 and 1.0).

[Figure]

**Figure S2.** Same as Figure S1 but with Pearson correlation coefficients which denoted by CORR.

**Comment 2:** Please use a larger fontsize for Figure 10 and 11. The caption tells "the increments in the RMSEs of the surface PM2.5 forecasts (g, h, i) and the increments in the RMSEs of the surface PM2.5 forecast (j, k, i)". Something goes wrong there?

**Reply:** We agree with the comment. The size of font in Figure 10 and Figure 11 is small as well as the dpi, and these two figures have been revised. "increments in the" in the caption is superfluous and has been removed.

**Comment 3:** Taking Figure 14 and 15, the authors found that the system seems not well constrained by DA in high level. Could the authors add some discussions about the physical reason or other aspects behind this problem?

Reply: Atmospheric chemistry-transport model (CTM) is an approximate representation of the evolution of air pollutants, which contains several physical and chemical processes such as direct emission, advection, diffusion, dry deposition, wet deposition, aqueous chemistry, gas-phase chemistry, heterogeneous chemical processes and so on. The concentration variability of gases and aerosols are not only affected by the above processes and are also constrained by meteorological input, initial conditions and boundary conditions. Emission is one of the most significant uncertainty sources. Studies on the analysis and forecast of air pollutants with CTM usually perturbed the initial emission to create initial ensemble (Tang et al., 2011; Kong et al., 2020; Dai et al., 2019; Cheng et al., 2019). The inversion estimation of emissions with CTM data assimilation is even a research hotspot (Kong et al., 2019; Wu et al., 2020; Feng et al., 2020; Tang et al., 2013; Ma et al., 2019; Dai et al., 2021), which is not the focus in our work.

Emission which is one of input of CTM can be divided into emission from agriculture, biomass burning, industry, power plant, resident, transportation. However, the most kinds of emissions mainly concentrated around the surface. Only biogenic emission, industrial emission and emission from power plant can emit air pollutants at a certain altitude. As a result, after perturbing initial emission to create initial ensemble, the error character (represented by the ensemble spread) of extinction coefficient

profiles on the background simulations with a clear decreasing from the surface to a certain altitude (blue curve in Figure 15a). It means that the analysis increment of each assimilation cycle tends to apportion more aerosol concentration (which can transform to extinction coefficient with observation operator) near surface. Therefore, the significant adjustment of aerosol extinction coefficients mainly occurs below the altitude of 3 km (Figure 14). The averaged extinction profiles (red curve in Figure 14) show a maximum value around the altitude of 600 m, which is in the planetary boundary layer (PBL). The most polluted plume occurs in PBL (usually 1 ~ 2 km) and affects the concentration of surface $PM_{2.5}$ (Yang et al., 2010; Lei et al., 2021).

In summary, the limited altitude of emission which is the perturbed to create ensemble limits the constraint of DA in high level. However, it is well constrained the aerosol profiles in the PBL which can significantly affect the surface.

**References:**

Cheng, Y., Dai, T., Goto, D., Schutgens, N. A. J., Shi, G., and Nakajima, T.: Investigating the assimilation of CALIPSO global aerosol vertical observations using a four-dimensional ensemble Kalman filter, Atmos. Chem. Phys., 19, 13445–13467, https://doi.org/10.5194/acp-19-13445-2019, 2019.

Dai, T., Cheng, Y., Suzuki, K., Goto, D., Kikuchi, M., Schutgens, N. A. J, Yoshida, M., Zhang, P., Husi, L., Shi, G., and Nakajima, T.: Hourly Aerosol Assimilation of Himawari-8 AOT Using the Four-Dimensional Local Ensemble Transform Kalman Filter, J. Adv. Model. Earth Syst., 11, 680–711, https://doi.org/10.1029/2018MS001475, 2019.

Dai, T., Cheng, Y., Goto, D., Li, Y., Tang, X., Shi, G., and Nakajima, T.: Revealing the sulfur dioxide emission reductions in China by assimilating surface observations in WRF-Chem, Atmos. Chem. Phys., 21, 4357–4379, https://doi.org/10.5194/acp-21-4357-2021, 2021.

Feng, S., Jiang, F., Wu, Z., Wang, H., Ju, W., and Wang, H.: CO Emissions Inferred From Surface CO Observations Over China in December 2013 and 2017, J. Geophys. Res. Atmos., 125, https://doi.org/10.1029/2019JD031808, 2020.

Gillet-Chaulet, F.: Assimilation of surface observations in a transient marine ice sheet model using an ensemble Kalman filter, The Cryosphere, 14, 811–832, https://doi.org/10.5194/tc-14-811-2020, 2020.

Kong, L., Tang, X., Zhu, J., Wang, Z., Pan, Y., Wu, H., Wu, L., Wu, Q., He, Y., Tian, S., Xie, Y., Liu, Z., Sui, W., Han, L., and Carmichael, G.: Improved Inversion of Monthly Ammonia Emissions in China Based on the Chinese Ammonia Monitoring Network and Ensemble Kalman Filter, Environ. Sci. Technol., 53, 12529–12538, https://doi.org/10.1021/acs.est.9b02701, 2019.

Kong, L., Tang, X., Zhu, J., Wang, Z., Li, J., Wu, H., Wu, Q., Chen, H., Zhu, L., Wang, W., Liu, B., Wang, Q., Chen, D., Pan, Y., Song, T., Li, F., Zheng, H., Jia, G., Lu, M., Wu, L., and Carmichael, G. R.: A Six-year long (2013–2018) High-resolution Air Quality Reanalysis Dataset over China base on the assimilation of surface observations from CNEMC, Earth Syst. Sci. Data, 13, 529–570, https://doi.org/10.5194/essd-13-529-2021, 2020.

Lei, L., Sun, Y., Ouyang, B., Qiu, Y., Xie, C., Tang, G., Zhou, W., He, Y., Wang, Q., Cheng, X., Fu, P., and Wang, Z.: Vertical Distributions of Primary and Secondary Aerosols in Urban Boundary Layer: Insights into Sources, Chemistry, and Interaction with Meteorology, Environ. Sci. Technol., 55, 4542–4552, https://doi.org/10.1021/acs.est.1c00479, 2021.

Ma, C., Wang, T., Mizzi, A. P., Anderson, J. L., Zhuang, B., Xie, M., and Wu, R.:

Multiconstituent Data Assimilation With WRF-Chem/DART: Potential for Adjusting Anthropogenic Emissions and Improving Air Quality Forecasts Over Eastern China, J. Geophys. Res. Atmos., 2019JD030421, https://doi.org/10.1029/2019JD030421, 2019.

Ma, C., Wang, T., Jiang, Z., Wu, H., Zhao, M., Zhuang, B., Li, S., Xie, M., Li, M., Liu, J., and Wu, R.: Importance of Bias Correction in Data Assimilation of Multiple Observations Over Eastern China Using WRF-Chem/DART, J. Geophys. Res. Atmos., 125, https://doi.org/10.1029/2019JD031465, 2020.

Nerger, L.: On Serial Observation Processing in Localized Ensemble Kalman Filters, 143, 15, 2015.

Nerger, L.: Data assimilation for nonlinear systems with a hybrid nonlinear Kalman ensemble transform filter, Quart J Royal Meteoro Soc, qj.4221, https://doi.org/10.1002/qj.4221, 2021.

Pham, D. T., Verron, J., and Christine Roubaud, M.: A singular evolutive extended Kalman filter for data assimilation in oceanography, Journal of Marine Systems, 16, 323–340, https://doi.org/10.1016/S0924-7963(97)00109-7, 1998.

Tang, X., Zhu, J., Wang, Z. F., and Gbaguidi, A.: Improvement of ozone forecast over Beijing based on ensemble Kalman filter with simultaneous adjustment of initial conditions and emissions, Atmos. Chem. Phys., 11, 12901–12916, https://doi.org/10.5194/acp-11-12901-2011, 2011.

Tang, X., Zhu, J., Wang, Z. F., Wang, M., Gbaguidi, A., Li, J., Shao, M., Tang, G. Q., and Ji, D. S.: Inversion of CO emissions over Beijing and its surrounding areas with ensemble Kalman filter, Atmospheric Environment, 81, 676–686, https://doi.org/10.1016/j.atmosenv.2013.08.051, 2013.

Wu, H., Tang, X., Wang, Z., Wu, L., Li, J., Wang, W., Yang, W., and Zhu, J.: High-spatiotemporal-resolution inverse estimation of CO and NOx emission reductions during emission control periods with a modified ensemble Kalman filter, Atmospheric Environment, 236, 117631, https://doi.org/10.1016/j.atmosenv.2020.117631, 2020.

Yang, T., Wang, Z., Zhang, B., Wang, X., Wang, W., Gbauidi, A., and Gong, Y.: Evaluation of the effect of air pollution control during the Beijing 2008 Olympic Games using Lidar data, Chin. Sci. Bull., 55, 1311–1316, https://doi.org/10.1007/s11434-010-0081-y, 2010.

---

## Author Comment (AC2)

The authors appreciate the reviewers for the effort to review our manuscript and to provide constructive comments. As suggested, we carefully revised the manuscript thoroughly according to the valuable advices. Listed below are our point-by-point responses in blue to the reviewer's comments in black. The reviewer's comments are reproduced (black) along with our replies and changes made to the text in the revised manuscript.

**Anonymous Referee #2**

**General comments:** Aerosol vertical structure is important for investigating global climate change, air pollution transport and control. The authors developed an online data assimilation system for vertical observation by coupling NAQPMS with PDAF, which offers spatiotemporally continuous aerosol vertical profiles. The system can make efficient use of parallel computational resources and produce great improvement in the aerosol vertical structure and surface PM2.5 concentration. Overall, the whole manuscript is within the scope of GMD and well written. I think the research is novelty to impact on other one's research and could be reconsidered after major revisions.

**Reply:** We thank the reviewer for the positive assessment and constructive suggestions of our manuscript.

**Comment 1:** In previous assimilation studies of satellite products, 6 hr or 12 hr has been chosen as the assimilation window. The whole manuscript is based the analysis and subsequent 1-hr forecast. So why do authors choose one hour as DA window? What is the difference between 1 hr and 6 hr or 12 or in terms of assimilation effect?

**Reply:** The assimilation window denotes the time length of the assimilation period (Wu et al., 2008). I guess the "assimilation window" in this comment actually means assimilation cycling. Therefore, we here reply on the setting of 1-hr as assimilation window and continuous 1-hr cycling, respectively.

Firstly, an EnKF system requires a short assimilation window in order to have the ensemble perturbations evolve linearly and remain Gaussian (Liu et al., 2019), which is different from 4D-Var which do require a long window reduce the effect of the

initially specified covariances (Pires et al., 1996). For the EnKF in general, it is desirable to use a short assimilation window (Houtekamer and Zhang, 2016). So we choose 1-hr as assimilation window in our EnKF system (NAQPMS-PDAF) as other similar studies do (Ma et al., 2020, 2019; Liu et al., 2019; Ha et al., 2020).

Secondly, the assimilation cycling is set as 1-hr in our work. On one hand, the main reason is that our manuscript focuses on investigating the parallel performance of NAQPMS-PDAF which is online coupled and the improvement of vertical profiles after assimilating aerosol extinction coefficient profile. The performance of ensemble forecast after ensemble filter of 1-hr or 6-hr is not the focus. Therefore, we increase the frequency of assimilation from every 6-hr to 1-hr. On the other hand, the 6-hr cycling usually choose 00:00, 06:00, 12:00 and 18:00 UTC as cycle times which is corresponding to the updated time of FNL data (Ma et al., 2020, 2019; Pang et al., 2018; Liu et al., 2011). However, this kind of setting may miss key values of aerosol extinction coefficients as extinction profiles measured by ground-based lidar has large temporal variability. Finally, the effect of forecast after assimilating ground-based lidar measurement is larger than 6-hr and even reach to 24-hr (Zheng, 2018).

We perform NP-LIDAR-6HR experiment shown in Table S1 to compare the assimilation effect between performing 1-hr and 6-hr cycling. Figure S1 and S2 are scatter plots and frequency distribution of extinction coefficients from the model versus the ground-based lidar measurements averaged over 5 DA sites and 6 VE sites of FR, NP-LIDAR and NP-LIDAR-6HR experiment, which are corresponding to Figure 8 and Figure 9 in our manuscript, respectively. As shown in Fig. S1f, extinction coefficient scatters are mainly concentrated around the 1:1 line in the NP-LIDAR-6HR experiment at DA sites. The RMSE and CORR value decreases (increases) from 0.42 1/km (0.33) in the FR experiment to 0.18 1/km (0.89) in the NP-LIDAR-6HR experiment, showing that the effect of assimilating lidar measurement with 6-hr cycling is positive. As shown in Fig. S1e and Fig. S1f, the RMSE value of the NP-LIDAR and NP-LIDAR-6HR experiment is 0.16 1/km and 0.18 1/km, respectively. The CORR value of these two experiments is 0.91 and 0.89. Note that the analysis time period in the NP-LIDAR-6HR is almost 6 times shorter than that in the NP-LIDAR experiment, result in a smaller

number of scatters in Fig. S1f than that in Fig. S1e. Fig. S2c and Fig. S2d also show that the performance of BIAS of the NP-LIDAR experiment is slightly better than that of the NP-LIDAR-6HR experiment with 93 % and 92 % scatters within |BIAS| < 0.25. It can be found that the statistic performance of the NP-LIDAR experiment is close to that in the NP-LIDAR-6HR experiment, and the performance of the former is slightly better than that in the latter. It means that the performance of assimilating all lidar measurements with 1-hr cycling is slightly better than assimilating the lidar measurements with 6-hr cycling under the current configuration.

As shown in Fig. S1b and Fig. S1c, the RMSE (CORR) value is 0.27 1/km (0.72) and 0.33 1/km (0.60) in the NP-LIDAR and NP-LIDAR-6HR experiment at DA sites, respectively. The frequency of |BIAS| <0.25 is 80 % and 75 % in the NP-LIDAR and NP-LIDAR-6HR experiments at DA sites, which is shown in Fig. S2a and Fig. S2b. It indicates that the statistic performance of the 1-hr forecast in the NP-LIDAR is better than that in the NP-LIDAR-6HR. It can be explained that the performance of NP-LIDAR is much less affected by the attenuation of data assimilation due to 1-hr is less than 6-hr in the NP-LIDAR-6HR experiment. At the VE sites, the statistic performance of extinction coefficients in the NP-LIDAR (Fig. S1h and Fig. S2e) and NP-LIDAR-6HR (Fig. S1i and Fig. S2f) experiment is nearly close, which both show a significantly improvement than that in the FR experiment (Fig. S1g).

**Table S1.** Summary of the Experimental design in AC2.

| Experiments | PM$_{2.5}$ DA | Ground-based lidar DA | DA cycling |
|---|---|---|---|
| FR | No | No | / |
| NP-LIDAR | No | Yes | 1-hr |
| NP-LIDAR-6HR | No | Yes | 6-hr |

[Figure]

**Figure S1.** Scatter plots of the modeled hourly extinction coefficients at 550 nm versus the ground lidar hourly aerosol extinction coefficients at 532 nm (1/km) of forecasts of FR (a)/(g), forecasts of NP-LIDAR(b)/(h), forecasts of NP-LIDAR-6HR (c)/(i), analysis of FR (d)/(j), analysis of NP-LIDAR (e)/(k), analysis of NP-LIDAR-6HR (f)/(l), which are averaged among DA sites/VE sites. The three dashed black lines correspond to the 1:2, 1:1 and 2:1 lines in each panel.

[Figure]

**Figure S2.** Frequency distributions of BIAS of forecasts of NP-LIDAR versus FR (a)/(e), forecasts of NP-LIDAR-6HR versus FR (b)/(f), analysis of NP-LIDAR versus FR (c)/(g) and analysis of NP-LIDAR-6HR versus FR (d)/(h), which are averaged among DA sites/VE sites.

**Comment 2:** The paper chooses ESTKF as the assimilation algorithm. What are the improvements or advantages of this Kalman filter algorithms compared to the previous KF algorithms?

Reply: The error-subspace transform Kalman filter (ESTKF, Nerger et al., 2012) is a recently developed Ensemble Kalman filter (EnKF, Evensen, 1994) variant which is derived from the singular evolutive interpolated Kalman filter (SEIK, Pham et al., 1998) by combining the advantages of the SEIK and the Ensemble Transform Kalman Filter (ETKF, Bishop et al., 2001).

Firstly, EnKF originated from the fusion of extended Kalman filter (EKF, Cohn, 1997) and Monte Carlo estimation methods. By providing flow- and location-dependent estimates of first-guess forecast error, the EnKF can potentially provide analysis and forecasts that are much more accurate than data assimilation schemes which assume that the background error does not vary in time (Whitaker and Hamill, 2002).

Secondly, EnKF and its variants can be categorized in deterministic ensemble filter, where the analysis is found through explicit mathematical transformations (SEIK, ETKF, ESTKF and so on), and stochastic ensemble filters, where perturbed forecasted observations are used (original EnKF). As one of ensemble square root filter algorithms, ESTKF is the former. On one hand, although stochastic filters can handle non-linearity better than the deterministic filters for large ensemble sizes (Lawson and Hansen, 2004), current computational resources restricts us to only use small ensemble sizes for high-dimensional problems. On the other hand, the stochastic filters may add another source of sampling error and underestimate the analysis update because observations assimilated is perturbed.

Thirdly, as mentioned above, the ESTKF is a combination of SEIK and ETKF and the three filters are essentially equivalent apart from computing the ensemble transformation in the error subspace (Vetra-Carvalho et al., 2018). The most significantly difference of ESTKF differs from SEIK and ETKF is that the error-subspace matrix is computed by

$$L = X^f \Omega, \tag{1}$$

where it is a projection matrix of size $N_e \times (N_e - 1)$ given by the set of equations as follows:

$$\Omega_{ij} = \begin{cases} 1 - \dfrac{1}{N_e}\dfrac{1}{\dfrac{1}{\sqrt{N_e}}+1} & \text{for } i=j, i<N_e \\[3ex] -\dfrac{1}{N_e}\dfrac{1}{\dfrac{1}{\sqrt{N_e}}+1} & \text{for } i \neq j, i<N_e \\[3ex] -\dfrac{1}{\sqrt{N_e}} & \text{for } i=N_e. \end{cases} \qquad (2)$$

where $N_e$ is the number of ensemble members and $i = 1, 2, ..., N_e$.

The ESTKF can exhibit better properties than the SEIK filter, like a minimum ensemble transformation as the ETKF (Vetra-Carvalho et al., 2018). Nerger et al. (2012) conducted a series of numerical experiments to compare the performance of SEIK, ETKF and ESTKF using deterministic and random ensemble transformations. They found that the performance for the ESTKF and ETKF are better than SEIK filter with ESTKF having a slightly lower computational cost.

**Comment 3:** What does "Although the orbits are slightly covered by the model domain, the only difference between the FR and NP-LIDAR experiment is whether ground-based lidar measurements are assimilated (Fig. 12b)" mean? I do not understand this very well. What is the connection between these two sentences?

**Reply:** We agree with the comment. This sentence is really ambiguous and has been revised.

**Changes in manuscript**: We have replaced the "Although the orbits are slightly covered by the model domain, the only difference between the FR and NP-LIDAR experiment is whether ground-based lidar measurements are assimilated (Fig. 12b)" with "Although the orbits are slightly covered by the model domain, the only difference of the averaged profiles between the FR and NP-LIDAR experiment is whether ground-based lidar measurements are assimilated (Fig. 12b)", please refer to the revised manuscript in Line 529-530.

**Comment 4:** L605: Adding "measured by lidar" after "The aerosol vertical profile" for clarity.

**Reply:** Thanks, we agree with this comment.

**Changes in manuscript:** We have replaced the "The aerosol vertical profile averaged over VE sites shows a similar shape to that over DA sites" with "The aerosol vertical profile from lidar measurements averaged over VE sites shows a similar shape to that over DA sites", please refer to the revised manuscript in Line 599.

**Comment 5:** L675: authors listed several reasons to explain that only assimilating lidar measurements has a weaker performance than only assimilating surface PM2.5 measurements. However, these reasons are just a guess without any detailed analysis. So, these reasons should not be listed in conclusion.

**Reply:** Thanks, we agree with this comment.

**Changes in manuscript:** We have deleted "This could be explained by the relatively sparser distribution of lidar sites compared with surface $PM_{2.5}$ measurements and the uncertainties in the spatial representation of lidar data, as well as the errors in the lumped variables of extinction coefficients with multiple contributions by different aerosol components. Moreover, the problem can also be attributed to the discordant relationship between aerosol mass concentration and extinction coefficients both in the simulation and measurements".

**Comment 6:** L685: "a systematic data quality control of lidar measurements is urgently needed to solve this problem in future research" should be deleted. The reason is same as the above comment.

**Reply:** Thanks, we agree with this comment.

**Changes in manuscript:** We have deleted "A systematic data quality control of lidar measurements is urgently needed to solve this problem in future research".

**Comment 7:** Fig. 7: The description of Fig. 7d is missing.

**Reply:** Thanks, it has been corrected in Line 1202.

**Changes in manuscript:** We have added "(d) extinction coefficients at 1000 m". Please refer to the revised manuscript in Line 1202.

**Comment 8:** Fig. 12: "(e)" is missing.

**Reply:** Thanks, it has been added. Please refer to the revised manuscript in Line 1241.

**Comment 9:** Fig. 15: "2021" should be "2019".

**Reply:** Thanks, it has been corrected in Line 1257 of the revised manuscript.

**References:**

Bishop, C. H., Etherton, B. J., and Majumdar, S. J.: Adaptive Sampling with the Ensemble Transform Kalman Filter. Part I: Theoretical Aspects, 129, 17, 2001.

Cohn, S. E.: An introduction to estimation theory, J. Meteor. Soc. Jap., 75, 257–288, 1997.

Evensen, G.: Sequential data assimilation with a nonlinear quasi-geostrophic model using Monte Carlo methods to forecast error statistics, J. Geophys. Res., 99, 10143, https://doi.org/10.1029/94JC00572, 1994.

Ha, S., Liu, Z., Sun, W., Lee, Y., and Chang, L.: Improving air quality forecasting with the assimilation of GOCI aerosol optical depth (AOD) retrievals during the KORUS-AQ period, 20, 6015–6036, https://doi.org/10.5194/acp-20-6015-2020, 2020.

Houtekamer, P. L. and Zhang, F.: Review of the Ensemble Kalman Filter for Atmospheric Data Assimilation, Mon. Wea. Rev., 144, 4489–4532, https://doi.org/10.1175/MWR-D-15-0440.1, 2016.

Lawson, W. G. and Hansen, J. A.: Implications of Stochastic and Deterministic Filters as Ensemble-Based Data Assimilation Methods in Varying Regimes of Error Growth, Mon. Wea. Rev., 132, 1966–1981, https://doi.org/10.1175/1520-0493(2004)132<1966:IOSADF>2.0.CO;2, 2004.

Liu, Y., Kalnay, E., Zeng, N., Asrar, G., Chen, Z., and Jia, B.: Estimating surface carbon fluxes based on a local ensemble transform Kalman filter with a short assimilation window and a long observation window: an observing system simulation experiment test in GEOS-Chem 10.1, Geosci. Model Dev., 12, 2899–2914, https://doi.org/10.5194/gmd-12-2899-2019, 2019.

Liu, Z., Liu, Q., Lin, H.-C., Schwartz, C. S., Lee, Y.-H., and Wang, T.: Three-dimensional variational assimilation of MODIS aerosol optical depth: Implementation and application to a dust storm over East Asia, 116, D23206, https://doi.org/10.1029/2011JD016159, 2011.

Ma, C., Wang, T., Mizzi, A. P., Anderson, J. L., Zhuang, B., Xie, M., and Wu, R.: Multiconstituent Data Assimilation With WRF-Chem/DART: Potential for Adjusting Anthropogenic Emissions and Improving Air Quality Forecasts Over Eastern China, J. Geophys. Res. Atmos., 2019JD030421, https://doi.org/10.1029/2019JD030421, 2019.

Ma, C., Wang, T., Jiang, Z., Wu, H., Zhao, M., Zhuang, B., Li, S., Xie, M., Li, M., Liu, J., and Wu, R.: Importance of Bias Correction in Data Assimilation of Multiple Observations Over Eastern China Using WRF-Chem/DART, J. Geophys. Res. Atmos., 125, https://doi.org/10.1029/2019JD031465, 2020.

Nerger, L., Janjić, T., Schröter, J., and Hiller, W.: A Unification of Ensemble Square

Root Kalman Filters, Mon. Wea. Rev., 140, 2335–2345, https://doi.org/10.1175/MWR-D-11-00102.1, 2012.

Pang, J., Liu, Z., Wang, X., Bresch, J., Ban, J., Cnen, D., and Kim, J.: Assimilating AOD retrievals from GOCI and VIIRS to forecast surface PM2.5 episodes over Eastern China, 179, 288–304, https://doi.org/10.1016/j.atmosenv.2018.02.011, 2018.

Pham, D. T., Verron, J., and Gourdeau, L.: Singular evolutive Kalman filters for data assimilation in oceanography, C. R. Acad. Sci. Ser. II, 326, 255–260, https://doi.org/10.1016/S1251-8050(97)86815-2, 1998.

Pires, C., Vautard, R., and Talagrand, O.: On extending the limits of variational assimilation in nonlinear chaotic systems, 48, 96–121, https://doi.org/10.1034/j.1600-0870.1996.00006.x, 1996.

Vetra-Carvalho, S., van Leeuwen, P. J., Nerger, L., Barth, A., Altaf, M. U., Brasseur, P., Kirchgessner, P., and Beckers, J.-M.: State-of-the-art stochastic data assimilation methods for high-dimensional non-Gaussian problems, Tellus A: Dynamic Meteorology and Oceanography, 70, 1–43, https://doi.org/10.1080/16000870.2018.1445364, 2018.

Whitaker, J. S. and Hamill, T. M.: Ensemble data assimilation without perturbed observations, Mon. Wea. Rev., 130, 1913–1924, https://doi.org/10.1175/MWR3156.1, 2002.

Wu, L., Mallet, V., Bocquet, M., and Sportisse, B.: A comparison study of data assimilation algorithms for ozone forecasts, J. Geophys. Res., 113, D20310, https://doi.org/10.1029/2008JD009991, 2008.

Zheng, H.: Improvement of PM2.5 Forecast by Data Assimilation of Ground and Lidar Observation, doctor, University of Science and Technology of China, 2018.

---

## Author Comment (AC3)

The authors appreciate the reviewers for the effort to review our manuscript and to provide constructive comments. As suggested, we carefully revised the manuscript thoroughly according to the valuable advices. Listed below are our point-by-point responses in blue to the reviewer's comments in black. The modifications corresponding to the comments and the revised language and grammars in the manuscript are marked in red.

**Anonymous Referee #1**

**General comments:** This paper describes a chemistry-transport model that is implemented with a data assimilation compartment using the Parallel Data Assimilation Framework. First of all, I would admit that I don't expertise in the atmosphere chemistry area, all my comments are from the data assimilation with PDAF. As for as I know, the online data assimilation approach with PDAF is first thoroughly described in Nerger et al., (2019) GMD paper, where the structure, the algorithm, the implementation are shown based on a climate model AWI-CM. Actually, such implementation has been widely used also for other ocean models such as FESOM, MITgcm etc. Another group like C-Coupler (Liu et al., 2020 GMD) provides similar data assimilation functions as well. I found this research is comprehensive with both technical and experimental perspectives. The results are clearly present and well organized. I would recommend acceptance after minor revisions. In general, I think the paper is well written with adequate evidence for their results.

**Reply:** We thank the reviewer for the positive assessment and constructive suggestions of our manuscript.

**Comment 1:** As for the paper structure, I would suggest the authors rephrase 2.3.1 and 2.3.2, i.e., the technical parts. Most of these implementation details are already well described in Nerger et al., 2019. I didn't see too many differences compared to Nerger's. Currently, it's rather a repetition. Please cite this paper directly and show your differences to condense the context. For 2.3.2, again, not necessarily a repetition of these algorithm details, which are well-known in amount of studies. The authors should concentrate on things that are distinguishable from others' work. For example,

localization radius, whether it is sensitive to your configuration (I see the authors cite two other researches, but actually these are different stories if the configuration changes based on my practices), are you using same localization radius for different observations; forgetting factor, which value is set, why is that; things like that.

**Reply:**

Firstly, we agree with that our manuscript should focus on the differences compared to other studies. The first part of Sec. 2.3.1 is the implementation of two level of parallelization based on MPI. We have cut out redundant descriptions but not deleted the content altogether because it is the first time to introduce PDAF into the study area of CTM. The second part of Sec. 2.3.1 is the main program flow. The third part of Sec. 2.3.1 is the description of the dimension of the state vectors used in PDAF which is specially redesigned in our work. The fourth part of Sec. 2.3.1 is two modules dealing with the data transfer before and after the time loop of the NAQPMS-PDAF which are also designed in our work. In Sec. 2.3.2, we have cited the literature about description of forecast step. The analysis step of ESTKF is retained because ESTKF is recently developed especially used in the area of CTM and the difference between ESTKF and other deterministic ensemble filter is mainly the computation of the ensemble transformation in the error subspace. Another anonymous reviewer wants more information about the PDAF and the ESTKF algorithm, so the revision above is the result of both considering two reviewers' comments.

Secondly, two kinds of observations (surface PM2.5 mass concentration and vertical profiles of aerosol extinction coefficients measured by ground-based lidar) are assimilated into NAQPMS-PDAF in this study. We set the localization radius as 200 km for both observations. For surface PM2.5 concentration, we follow Kong et al. (2020) and set the localization radius as 200 km, because the kind of observation, the atmospheric chemistry-transport model (NAQPMS) as well as the ensemble filter algorithm of this study is same as their work. Therefore, we here focus the localization radius of ground-based lidar and forgetting factor of the data assimilation system, which are set as 200 km and 0.96 in this study. The setting of forgetting factor is omitted in the original manuscript and have been revised.

The several sensitivity tests have been made to supplement the setting of these two data assimilation parameters in this study. We refer to Gillet-Chaulet (2020) about the period chosen for sensitivity tests when assimilating real observation under limited computational resources. For sensitivity tests, we choose the study period from 00:00 UTC 23 April to 05:00 UTC 23 April 2019 with abundant pollution plume measured by the ground-based lidar. A series of sensitivity tests are performed with the localization radius (5 km, 50 km, 100 km, 150 km, 200 km, 250 km, 300 km and 400 km) and forgetting factor (0.6, 0.7, 0.8, 0.9, 0.92, 0.94, 0.96, 0.98 and 1.0). The configuration of data assimilation is same as the NP-LIDAR-PM25 experiment in the manuscript expect for the study period. The results of sensitivity tests are evaluated by the VE sites (the ground-based lidar measurements not assimilated) of model domain. Following Nerger (2015) and Nerger (2021), Figure S1 and Figure S2 show the time-mean RMSE and Pearson correlation coefficient for the sensitivity tests, respectively. As we can see in Figure S1, RMSE of aerosol extinction coefficients converges for all combinations of localization radius and forgetting factor. The minimum RMSE of 0.36 1/km is obtained for localization radius of 200 km and forgetting factor of 0.96, 0.98 and 1.0. The maximum Pearson correlation coefficient of 0.75 is obtained for localization radius of 150 km and 200 km and is not sensitive to forgetting factor when forgetting factor is larger than 0.9. Forgetting factor is used to inflate the forecast covariance matrix to reduce under-sampling issues, especially in the long run (Pham et al., 1998). Although the statistical results vary slightly with forgetting factor due to relatively short run time, the combined results of RMSE and Pearson correlation coefficient can provide the optimal parameters in the series of sensitivity tests. To sum up, the forgetting factor of 0.96 and localization radius of ground-based lidar of 200 km is the most optimal parameters. Moreover, the setting of localization radius is same as Cheng et al. (2019) performing ensemble filter to assimilating lidar measurements, and is also close to Ma et al. (2020) performing ensemble filter to assimilating aerosol extinction coefficient profiles measured by ground-based lidar.

**Changes in manuscript**: Changes have been made in Sec. 2.3.1 in Line 176-183 of the revised manuscript and revised text is "As the NAQPMS described in Section

2.1 is well written and its source code is available, this study chooses the online method to couple the PDAF with the NAQPMS in order to gain the best performance. The core modification in the coupling is parallelization for ensemble simulations. Message Passing Interface standard (MPI; Gropp et al., 1994) both used in NAQPMS and PDAF allows each process to handle distributed parts of a program and data exchange. The communicator MPI_COMM_WORLD is used in NAQPMS as one-level parallelization to improve computational efficiency and the distribution of processes is exemplified in Fig. 1a.".

Also, changes have been made in Line 306-307 and revised text is "We set the horizontal localization radius and forgetting factor to 200 km and 0.96 according to a series of sensitivity tests in the Supplement and other related studies (Kong et al., 2020; Zhao et al., 2020; Cheng et al., 2019b; Ma et al., 2020)".

[Figure]

**Figure S1.** RMSE for sensitivity tests with localization radius of ground-based lidar (5 km, 50 km, 100 km, 150 km, 200 km, 250 km, 300 km and 400 km) and forgetting factor of NAQPMS-PDAF (0.6, 0.7, 0.8, 0.9, 0.92, 0.94, 0.96, 0.98 and 1.0).

[Figure]

**Figure S2.** Same as Figure S1 but with Pearson correlation coefficients which denoted by CORR.

**Comment 2:** Please use a larger fontsize for Figure 10 and 11. The caption tells "the increments in the RMSEs of the surface PM2.5 forecasts (g, h, i) and the increments in the RMSEs of the surface PM2.5 forecast (j, k, i)". Something goes wrong there?

**Reply:** We agree with the comment. The size of font in Figure 10 and Figure 11 is small as well as the dpi, and these two figures have been revised. "increments in the" in the caption is superfluous and has been removed.

**Changes in manuscript:** Changes have been made in Line 1291-1295 of the revised manuscript and revised text is "the RMSEs of the surface $PM_{2.5}$ forecasts (g, h, i) and the increments in the RMSEs of the surface $PM_{2.5}$ forecast (j, k, i)".

**Comment 3:** Taking Figure 14 and 15, the authors found that the system seems not well constrained by DA in high level. Could the authors add some discussions about the physical reason or other aspects behind this problem?

**Reply:** Atmospheric chemistry-transport model (CTM) is an approximate representation of the evolution of air pollutants, which contains several physical and chemical processes such as direct emission, advection, diffusion, dry deposition, wet deposition, aqueous chemistry, gas-phase chemistry, heterogeneous chemical processes and so on. The concentration variability of gases and aerosols are not only affected by the above processes and are also constrained by meteorological input, initial conditions and boundary conditions. Emission is one of the most significant uncertainty sources. Studies on the analysis and forecast of air pollutants with CTM usually perturbed the initial emission to create initial ensemble (Tang et al., 2011; Kong et al., 2020; Dai et al., 2019; Cheng et al., 2019). The inversion estimation of emissions with CTM data assimilation is even a research hotspot (Kong et al., 2019; Wu et al., 2020; Feng et al., 2020; Tang et al., 2013; Ma et al., 2019; Dai et al., 2021), which is not the focus in our work.

Emission which is one of input of CTM can be divided into emission from agriculture, biomass burning, industry, power plant, resident, transportation. However, the most kinds of emissions mainly concentrated around the surface. Only biogenic emission, industrial emission and emission from power plant can emit air pollutants at

a certain altitude. As a result, after perturbing initial emission to create initial ensemble, the error character (represented by the ensemble spread) of extinction coefficient profiles on the background simulations with a clear decreasing from the surface to a certain altitude (blue curve in Figure 15a). It means that the analysis increment of each assimilation cycle tends to apportion more aerosol concentration (which can transform to extinction coefficient with observation operator) near surface. Therefore, the significant adjustment of aerosol extinction coefficients mainly occurs below the altitude of 3 km (Figure 14). The averaged extinction profiles (red curve in Figure 14) show a maximum value around the altitude of 600 m, which is in the planetary boundary layer (PBL). The most polluted plume occurs in PBL (usually $1 \sim 2$ km) and affects the concentration of surface $PM_{2.5}$ (Yang et al., 2010; Lei et al., 2021).

In summary, the limited altitude of emission which is the perturbed to create ensemble limits the constraint of DA in high level. However, it is well constrained the aerosol profiles in the PBL which can significantly affect the surface.

**Changes in manuscript**: Changes have been made in Line 676-680 of the revised manuscript and revised text is "It means that the analysis increment of each assimilation cycle tends to apportion more aerosol concentration near the surface".

**References:**

Cheng, Y., Dai, T., Goto, D., Schutgens, N. A. J., Shi, G., and Nakajima, T.: Investigating the assimilation of CALIPSO global aerosol vertical observations using a four-dimensional ensemble Kalman filter, Atmos. Chem. Phys., 19, 13445–13467, https://doi.org/10.5194/acp-19-13445-2019, 2019.

Dai, T., Cheng, Y., Suzuki, K., Goto, D., Kikuchi, M., Schutgens, N. A. J., Yoshida, M., Zhang, P., Husi, L., Shi, G., and Nakajima, T.: Hourly Aerosol Assimilation of Himawari-8 AOT Using the Four-Dimensional Local Ensemble Transform Kalman Filter, J. Adv. Model. Earth Syst., 11, 680–711, https://doi.org/10.1029/2018MS001475, 2019.

Dai, T., Cheng, Y., Goto, D., Li, Y., Tang, X., Shi, G., and Nakajima, T.: Revealing the sulfur dioxide emission reductions in China by assimilating surface observations in WRF-Chem, Atmos. Chem. Phys., 21, 4357–4379, https://doi.org/10.5194/acp-21-4357-2021, 2021.

Feng, S., Jiang, F., Wu, Z., Wang, H., Ju, W., and Wang, H.: CO Emissions Inferred From Surface CO Observations Over China in December 2013 and 2017, J. Geophys. Res. Atmos., 125, https://doi.org/10.1029/2019JD031808, 2020.

Gillet-Chaulet, F.: Assimilation of surface observations in a transient marine ice sheet model using an ensemble Kalman filter, The Cryosphere, 14, 811–832, https://doi.org/10.5194/tc-14-811-2020, 2020.

Kong, L., Tang, X., Zhu, J., Wang, Z., Pan, Y., Wu, H., Wu, L., Wu, Q., He, Y., Tian, S., Xie, Y., Liu, Z., Sui, W., Han, L., and Carmichael, G.: Improved Inversion of Monthly Ammonia Emissions in China Based on the Chinese Ammonia Monitoring Network and Ensemble Kalman Filter, Environ. Sci. Technol., 53, 12529–12538, https://doi.org/10.1021/acs.est.9b02701, 2019.

Kong, L., Tang, X., Zhu, J., Wang, Z., Li, J., Wu, H., Wu, Q., Chen, H., Zhu, L., Wang, W., Liu, B., Wang, Q., Chen, D., Pan, Y., Song, T., Li, F., Zheng, H., Jia, G., Lu, M., Wu, L., and Carmichael, G. R.: A Six-year long (2013–2018) High-resolution Air Quality Reanalysis Dataset over China base on the assimilation of surface observations from CNEMC, Earth Syst. Sci. Data, 13, 529–570, https://doi.org/10.5194/essd-13-529-2021, 2020.

Lei, L., Sun, Y., Ouyang, B., Qiu, Y., Xie, C., Tang, G., Zhou, W., He, Y., Wang, Q., Cheng, X., Fu, P., and Wang, Z.: Vertical Distributions of Primary and Secondary Aerosols in Urban Boundary Layer: Insights into Sources, Chemistry, and Interaction with Meteorology, Environ. Sci. Technol., 55, 4542–4552, https://doi.org/10.1021/acs.est.1c00479, 2021.

Ma, C., Wang, T., Mizzi, A. P., Anderson, J. L., Zhuang, B., Xie, M., and Wu, R.:

Multiconstituent Data Assimilation With WRF-Chem/DART: Potential for Adjusting Anthropogenic Emissions and Improving Air Quality Forecasts Over Eastern China, J. Geophys. Res. Atmos., 2019JD030421, https://doi.org/10.1029/2019JD030421, 2019.

Ma, C., Wang, T., Jiang, Z., Wu, H., Zhao, M., Zhuang, B., Li, S., Xie, M., Li, M., Liu, J., and Wu, R.: Importance of Bias Correction in Data Assimilation of Multiple Observations Over Eastern China Using WRF-Chem/DART, J. Geophys. Res. Atmos., 125, https://doi.org/10.1029/2019JD031465, 2020.

Nerger, L.: On Serial Observation Processing in Localized Ensemble Kalman Filters, 143, 15, 2015.

Nerger, L.: Data assimilation for nonlinear systems with a hybrid nonlinear Kalman ensemble transform filter, Quart J Royal Meteoro Soc, qj.4221, https://doi.org/10.1002/qj.4221, 2021.

Pham, D. T., Verron, J., and Christine Roubaud, M.: A singular evolutive extended Kalman filter for data assimilation in oceanography, Journal of Marine Systems, 16, 323–340, https://doi.org/10.1016/S0924-7963(97)00109-7, 1998.

Tang, X., Zhu, J., Wang, Z. F., and Gbaguidi, A.: Improvement of ozone forecast over Beijing based on ensemble Kalman filter with simultaneous adjustment of initial conditions and emissions, Atmos. Chem. Phys., 11, 12901–12916, https://doi.org/10.5194/acp-11-12901-2011, 2011.

Tang, X., Zhu, J., Wang, Z. F., Wang, M., Gbaguidi, A., Li, J., Shao, M., Tang, G. Q., and Ji, D. S.: Inversion of CO emissions over Beijing and its surrounding areas with ensemble Kalman filter, Atmospheric Environment, 81, 676–686, https://doi.org/10.1016/j.atmosenv.2013.08.051, 2013.

Wu, H., Tang, X., Wang, Z., Wu, L., Li, J., Wang, W., Yang, W., and Zhu, J.: High-spatiotemporal-resolution inverse estimation of CO and NOx emission reductions during emission control periods with a modified ensemble Kalman filter, Atmospheric Environment, 236, 117631, https://doi.org/10.1016/j.atmosenv.2020.117631, 2020.

Yang, T., Wang, Z., Zhang, B., Wang, X., Wang, W., Gbauidi, A., and Gong, Y.: Evaluation of the effect of air pollution control during the Beijing 2008 Olympic Games using Lidar data, Chin. Sci. Bull., 55, 1311–1316, https://doi.org/10.1007/s11434-010-0081-y, 2010.

---

## Author Comment (AC4)

The authors appreciate the reviewers for the effort to review our manuscript and to provide constructive comments. As suggested, we carefully revised the manuscript thoroughly according to the valuable advices. Listed below are our point-by-point responses in blue to the reviewer's comments in black. The reviewer's comments are reproduced (black) along with our replies and changes made to the text in the revised manuscript.

**Anonymous Referee #2**

**General comments:** Aerosol vertical structure is important for investigating global climate change, air pollution transport and control. The authors developed an online data assimilation system for vertical observation by coupling NAQPMS with PDAF, which offers spatiotemporally continuous aerosol vertical profiles. The system can make efficient use of parallel computational resources and produce great improvement in the aerosol vertical structure and surface PM2.5 concentration. Overall, the whole manuscript is within the scope of GMD and well written. I think the research is novelty to impact on other one's research and could be reconsidered after major revisions.

**Reply:** We thank the reviewer for the positive assessment and constructive suggestions of our manuscript.

**Comment 1:** In previous assimilation studies of satellite products, 6 hr or 12 hr has been chosen as the assimilation window. The whole manuscript is based the analysis and subsequent 1-hr forecast. So why do authors choose one hour as DA window? What is the difference between 1 hr and 6 hr or 12 or in terms of assimilation effect?

**Reply:** The assimilation window denotes the time length of the assimilation period (Wu et al., 2008). I guess the "assimilation window" in this comment actually means assimilation cycling. Therefore, we here reply on the setting of 1-hr as assimilation window and continuous 1-hr cycling, respectively.

Firstly, the EnKF system used in our work provides possibilities for using a short assimilation window to have the ensemble perturbations evolve linearly (Houtekamer and Zhang, 2016; Liu et al., 2019), while a 4D-Var system needs to keep a long window

to reduce the effect of the initially specified covariances (Pires et al., 1996). So we choose 1-hr as assimilation window in our EnKF system (NAQPMS-PDAF) as other similar studies do (Ma et al., 2020, 2019; Liu et al., 2019; Ha et al., 2020).

Secondly, the assimilation cycling is set as 1-hr in our work. On one hand, the main reason is that our manuscript focuses on investigating the parallel performance of NAQPMS-PDAF which is online coupled and the improvement of vertical profiles after assimilating aerosol extinction coefficient profile. The performance of ensemble forecast after ensemble filter of 1-hr or 6-hr is not the focus. Therefore, we increase the frequency of assimilation from every 6-hr to 1-hr. On the other hand, the 6-hr assimilation cycle used in similar studies (Ma et al., 2020, 2019; Pang et al., 2018; Liu et al., 2011) follows the model configuration of assimilating satellite data with coarse temporal resolution. However, the lidar measurements used in our work can provide large temporal variability with temporal resolution of 1-hr.

Therefore, we perform NP-LIDAR-6HR experiment shown in Table S1 to compare the assimilation effect between performing 1-hr and 6-hr cycling. Figure S1 and S2 are scatter plots and frequency distribution of extinction coefficients from the model versus the ground-based lidar measurements averaged over 5 DA sites and 6 VE sites of FR, NP-LIDAR and NP-LIDAR-6HR experiment, which are corresponding to Figure 8 and Figure 9 in our manuscript, respectively. As shown in Fig. S1f, extinction coefficient scatters are mainly concentrated around the 1:1 line in the NP-LIDAR-6HR experiment at DA sites. The RMSE and CORR value decreases (increases) from 0.42 1/km (0.33) in the FR experiment to 0.18 1/km (0.89) in the NP-LIDAR-6HR experiment, showing that the effect of assimilating lidar measurement with 6-hr cycling is positive. As shown in Fig. S1e and Fig. S1f, the RMSE value of the NP-LIDAR and NP-LIDAR-6HR experiment is 0.16 1/km and 0.18 1/km, respectively. The CORR value of these two experiments is 0.91 and 0.89. Note that the analysis time period in the NP-LIDAR-6HR is almost 6 times shorter than that in the NP-LIDAR experiment, result in a smaller number of scatters in Fig. S1f than that in Fig. S1e. Fig. S2c and Fig. S2d also show that the performance of BIAS of the NP-LIDAR experiment is slightly better than that of the NP-LIDAR-6HR experiment with 93 % and 92 % scatters within

|BIAS| < 0.25. It can be found that the statistic performance of the NP-LIDAR experiment is close to that in the NP-LIDAR-6HR experiment, and the performance of the former is slightly better than that in the latter. It means that the performance of assimilating all lidar measurements with 1-hr cycling is slightly better than assimilating the lidar measurements with 6-hr cycling under the current configuration.

As shown in Fig. S1b and Fig. S1c, the RMSE (CORR) value is 0.27 1/km (0.72) and 0.33 1/km (0.60) in the NP-LIDAR and NP-LIDAR-6HR experiment at DA sites, respectively. The frequency of |BIAS| <0.25 is 80 % and 75 % in the NP-LIDAR and NP-LIDAR-6HR experiments at DA sites, which is shown in Fig. S2a and Fig. S2b. It indicates that the statistic performance of the 1-hr forecast in the NP-LIDAR is better than that in the NP-LIDAR-6HR. It can be explained that the performance of NP-LIDAR is much less affected by the attenuation of data assimilation due to 1-hr is less than 6-hr in the NP-LIDAR-6HR experiment. At the VE sites, the statistic performance of extinction coefficients in the NP-LIDAR (Fig. S1h and Fig. S2e) and NP-LIDAR-6HR (Fig. S1i and Fig. S2f) experiment is nearly close, which both show a significantly improvement than that in the FR experiment (Fig. S1g).

**Changes in manuscript**: Changes have been made in Line 375-390 of the revised manuscript and revised text is "The EnKF system used in our work provides possibilities for using a short assimilation window to have the ensemble perturbations evolve linearly (Houtekamer and Zhang, 2016; Liu et al., 2019b), while a 4D-Var system needs to keep a long window to reduce the effect of the initially specified covariances (Pires et al., 1996). Therefore, we choose 1-hr as assimilation window in NAQPMS-PDAF as other similar studies (Liu et al., 2019b; Ma et al., 2019; Ha et al., 2020) do. The assimilation cycle is set as 1-hr. On one hand, our work focuses on investigating the parallel performance of NAQPMS-PDAF and the improvement of vertical profile simulations after assimilating aerosol extinction coefficient profiles. The performance of ensemble forecast after ensemble filter is not the focus. On the other hand, the 6-hr assimilation cycle used in similar studies (Liu et al., 2011; Ma et al., 2019, 2020; Pang et al., 2018) follows the model configuration of assimilating satellite data with coarse temporal resolution. However, the lidar measurements used in

our work can provide large temporal variability with temporal resolution of 1-hr. In order to investigate the difference of different assimilation cycle on the analysis and forecast, an extra experiment assimilating lidar measurements at cycle of 6-hr (NP-LIDAR-6HR) has been performed in the Supplement (Table S1). The RMSE value of the NP-LIDAR and NP-LIDAR-6HR experiment is 0.16 1/km and 0.18 1/km, respectively (Fig. S1). The CORR values of these two experiments is 0.91 and 0.89 (Fig. S1). The performance of BIAS of the NP-LIDAR experiment is slightly better than that of the NP-LIDAR-6HR experiment with 93 % and 92 % scatters within |BIAS| < 0.25. Other detailed discussion can be found in the Supplement. It can be found that the statistic performance at cycle of 1-hr is better than that at cycle with 6-hr, which supports the setting of cycle of 1-hr in our work.".

**Table S1.** Summary of the Experimental design in AC2.

| Experiments | $PM_{2.5}$ DA | Ground-based lidar DA | DA cycling |
|---|---|---|---|
| FR | No | No | / |
| NP-LIDAR | No | Yes | 1-hr |
| NP-LIDAR-6HR | No | Yes | 6-hr |

[Figure]

**Figure S1.** Scatter plots of the modeled hourly extinction coefficients at 550 nm versus the ground lidar hourly aerosol extinction coefficients at 532 nm (1/km) of forecasts of FR (a)/(g), forecasts of NP-LIDAR(b)/(h), forecasts of NP-LIDAR-6HR (c)/(i), analysis of FR (d)/(j), analysis of NP-LIDAR (e)/(k), analysis of NP-LIDAR-6HR (f)/(l), which are averaged among DA sites/VE sites. The three dashed black lines correspond to the 1:2, 1:1 and 2:1 lines in each panel.

[Figure]

**Figure S2.** Frequency distributions of BIAS of forecasts of NP-LIDAR versus FR (a)/(e), forecasts of NP-LIDAR-6HR versus FR (b)/(f), analysis of NP-LIDAR versus FR (c)/(g) and analysis of NP-LIDAR-6HR versus FR (d)/(h), which are averaged among DA sites/VE sites.

**Comment 2:** The paper chooses ESTKF as the assimilation algorithm. What are the improvements or advantages of this Kalman filter algorithms compared to the previous KF algorithms?

**Reply:**

Firstly, the error-subspace transform Kalman filter (ESTKF, Nerger et al., 2012) is a recently developed Ensemble Kalman filter (EnKF, Evensen, 1994) variant. EnKF originated from the fusion of extended Kalman filter (EKF, Cohn, 1997) and Monte Carlo estimation methods. By providing flow- and location-dependent estimates of first-guess forecast error, the EnKF can potentially provide analysis and forecasts that are much more accurate than data assimilation schemes which assume that the background error does not vary in time (Whitaker and Hamill, 2002).

Secondly, EnKF and its variants can be categorized in deterministic ensemble filter, where the analysis is found through explicit mathematical transformations (SEIK, ETKF, ESTKF and so on), and stochastic ensemble filters, where perturbed forecasted observations are used (original EnKF). As one of ensemble square root filter algorithms, ESTKF is the former. On one hand, the deterministic ensemble filter can only use small ensemble sizes for high-dimensional problems, while stochastic filters need large ensemble sizes (Lawson and Hansen, 2004). On the other hand, the stochastic filters may add another source of sampling error and underestimate the analysis update because observations assimilated is perturbed.

Thirdly, ESTKF is derived from the singular evolutive interpolated Kalman filter (SEIK, Pham et al., 1998) by combining the advantages of the SEIK and the Ensemble Transform Kalman Filter (ETKF, Bishop et al., 2001). These three filters are essentially equivalent apart from computing the ensemble transformation in the error subspace (Vetra-Carvalho et al., 2018). The most significantly difference of ESTKF differs from SEIK and ETKF is that the error-subspace matrix is computed by

$$L = X^f \Omega, \tag{1}$$

where it is a projection matrix of size $N_e \times (N_e - 1)$ given by the set of equations as follows:

$$\Omega_{ij} = \begin{cases} 1 - \dfrac{1}{N_e} \dfrac{1}{\dfrac{1}{\sqrt{N_e}} + 1} & \text{for } i = j, i < N_e \\[4ex] -\dfrac{1}{N_e} \dfrac{1}{\dfrac{1}{\sqrt{N_e}} + 1} & \text{for } i \neq j, i < N_e \\[4ex] -\dfrac{1}{\sqrt{N_e}} & \text{for } i = N_e. \end{cases} \qquad (2)$$

where $N_e$ is the number of ensemble members and $i = 1, 2, ..., N_e$.

The ESTKF can exhibit better properties than the SEIK filter, like a minimum ensemble transformation as the ETKF (Vetra-Carvalho et al., 2018). Nerger et al. (2012) conducted a series of numerical experiments to compare the performance of SEIK, ETKF and ESTKF using deterministic and random ensemble transformations. They found that the performance for the ESTKF and ETKF are better than SEIK filter with ESTKF having a slightly lower computational cost.

**Changes in manuscript**: Changes have been made in Line 252-266 of the revised manuscript and revised text is "The error subspace transform Kalman filter (ESTKF, Nerger et al., 2012) used in this study is a recently developed EnKF variant. Firstly, EnKF originated from the fusion of Kalman filter theory and Monte Carlo estimation method. By providing flow-dependent estimates of first-guess forecast error, the EnKF can potentially provide analysis and forecasts that are much more accurate than DA schemes which assume that the background error does not vary in time (Whitaker and Hamill, 2002). Secondly, EnKF and its variants can be categorized in deterministic filter (ETKF, ESTKF and so on) and stochastic filter which assimilates perturbed observations (original EnKF). On one hand, the deterministic filter can only use small ensemble sizes for high-dimensional problems, while stochastic filters need large ensemble sizes (Lawson and Hansen, 2004). On the other hand, the stochastic filters may add another source of sampling error and underestimate the analysis update because observations assimilated is perturbed. Thirdly, ESTKF is derived from the singular evolutive interpolated Kalman filter (SEIK, Pham et al., 1998) by combining the advantages of the SEIK and ETKF.These three filters are essentially equivalent apart from computing the ensemble transformation in the error subspace (Vetra-

Carvalho et al., 2018). The ESTKF can exhibit better properties than the SEIK filter, like a minimum ensemble transformation as the ETKF. Nerger et al. (2012) conducted a series of numerical experiments to compare the performance of SEIK, ETKF and ESTKF using deterministic and random ensemble transformations. They found that the performance for the ESTKF and ETKF are better than SEIK filter with ESTKF having a slightly lower computational cost. The ESTKF is outlined in this section." .

**Comment 3:** What does "Although the orbits are slightly covered by the model domain, the only difference between the FR and NP-LIDAR experiment is whether ground-based lidar measurements are assimilated (Fig. 12b)" mean? I do not understand this very well. What is the connection between these two sentences?

**Reply:** We agree with the comment. This sentence is really ambiguous and has been revised.

**Changes in manuscript**: Changes have been made in Line 572-573 of the revised manuscript and revised text is "Although the orbits are slightly covered by the model domain, the only difference of the averaged profiles between the FR and NP-LIDAR experiment is whether ground-based lidar measurements are assimilated (Fig. 12b)".

**Comment 4:** L605: Adding "measured by lidar" after "The aerosol vertical profile" for clarity.

**Reply:** Thanks, we agree with this comment.

**Changes in manuscript:** Changes have been made in Line 642 of the revised manuscript and revised text is "The aerosol vertical profile from lidar measurements averaged over VE sites shows a similar shape to that over DA sites".

**Comment 5:** L675: authors listed several reasons to explain that only assimilating lidar measurements has a weaker performance than only assimilating surface PM2.5 measurements. However, these reasons are just a guess without any detailed analysis. So, these reasons should not be listed in conclusion.

**Reply:** Thanks, we agree with this comment.

**Changes in manuscript:** We have deleted "This could be explained by the relatively sparser distribution of lidar sites compared with surface $PM_{2.5}$ measurements and the uncertainties in the spatial representation of lidar data, as well as the errors in the lumped variables of extinction coefficients with multiple contributions by different aerosol components. Moreover, the problem can also be attributed to the discordant

relationship between aerosol mass concentration and extinction coefficients both in the simulation and measurements". Please refer to the revised manuscript in Line 713-717.

**Comment 6:** L685: "a systematic data quality control of lidar measurements is urgently needed to solve this problem in future research" should be deleted. The reason is same as the above comment.

**Reply:** Thanks, we agree with this comment.

**Changes in manuscript:** We have deleted "A systematic data quality control of lidar measurements is urgently needed to solve this problem in future research". Please refer to the revised manuscript in Line 726-727.

**Comment 7:** Fig. 7: The description of Fig. 7d is missing.

**Reply:** Thanks, it has been corrected in Line 1272.

**Changes in manuscript:** Changes have been made in Line 1272 of the revised manuscript and revised text is "Time series of prior RMSE and total spread over all observations for (a) extinction coefficients at 50 m, (b) extinction coefficients at 150 m, (c) extinction coefficients at 502 m, (d) extinction coefficients at 1000 m and (d) the surface PM$_{2.5}$".

**Comment 8:** Fig. 12: "(e)" is missing.

**Reply:** Thanks, it has been added.

**Changes in manuscript:** Changes have been revised in Line 1312 of the revised manuscript and revised text is "05:00 UTC 18 April 2019 (e)".

**Comment 9:** Fig. 15: "2021" should be "2019".

**Reply:** Thanks, it has been corrected in Line 1328 of the revised manuscript.

**Changes in manuscript:** Changes have been made in Line 1328 of the revised manuscript and revised text is "All results are averaged over 1-30 April 2019".

**References:**

Bishop, C. H., Etherton, B. J., and Majumdar, S. J.: Adaptive Sampling with the Ensemble Transform Kalman Filter. Part I: Theoretical Aspects, 129, 17, 2001.

Cohn, S. E.: An introduction to estimation theory, J. Meteor. Soc. Jap., 75, 257–288, 1997.

Evensen, G.: Sequential data assimilation with a nonlinear quasi-geostrophic model using Monte Carlo methods to forecast error statistics, J. Geophys. Res., 99, 10143, https://doi.org/10.1029/94JC00572, 1994.

Ha, S., Liu, Z., Sun, W., Lee, Y., and Chang, L.: Improving air quality forecasting with the assimilation of GOCI aerosol optical depth (AOD) retrievals during the KORUS-AQ period, 20, 6015–6036, https://doi.org/10.5194/acp-20-6015-2020, 2020.

Houtekamer, P. L. and Zhang, F.: Review of the Ensemble Kalman Filter for Atmospheric Data Assimilation, Mon. Wea. Rev., 144, 4489–4532, https://doi.org/10.1175/MWR-D-15-0440.1, 2016.

Lawson, W. G. and Hansen, J. A.: Implications of Stochastic and Deterministic Filters as Ensemble-Based Data Assimilation Methods in Varying Regimes of Error Growth, Mon. Wea. Rev., 132, 1966–1981, https://doi.org/10.1175/1520-0493(2004)132<1966:IOSADF>2.0.CO;2, 2004.

Liu, Y., Kalnay, E., Zeng, N., Asrar, G., Chen, Z., and Jia, B.: Estimating surface carbon fluxes based on a local ensemble transform Kalman filter with a short assimilation window and a long observation window: an observing system simulation experiment test in GEOS-Chem 10.1, Geosci. Model Dev., 12, 2899–2914, https://doi.org/10.5194/gmd-12-2899-2019, 2019.

Liu, Z., Liu, Q., Lin, H.-C., Schwartz, C. S., Lee, Y.-H., and Wang, T.: Three-dimensional variational assimilation of MODIS aerosol optical depth: Implementation and application to a dust storm over East Asia, 116, D23206, https://doi.org/10.1029/2011JD016159, 2011.

Ma, C., Wang, T., Mizzi, A. P., Anderson, J. L., Zhuang, B., Xie, M., and Wu, R.: Multiconstituent Data Assimilation With WRF-Chem/DART: Potential for Adjusting Anthropogenic Emissions and Improving Air Quality Forecasts Over Eastern China, J. Geophys. Res. Atmos., 2019JD030421, https://doi.org/10.1029/2019JD030421, 2019.

Ma, C., Wang, T., Jiang, Z., Wu, H., Zhao, M., Zhuang, B., Li, S., Xie, M., Li, M., Liu, J., and Wu, R.: Importance of Bias Correction in Data Assimilation of Multiple Observations Over Eastern China Using WRF-Chem/DART, J. Geophys. Res. Atmos., 125, https://doi.org/10.1029/2019JD031465, 2020.

Nerger, L., Janjić, T., Schröter, J., and Hiller, W.: A Unification of Ensemble Square Root Kalman Filters, Mon. Wea. Rev., 140, 2335–2345, https://doi.org/10.1175/MWR-D-11-00102.1, 2012.

Pang, J., Liu, Z., Wang, X., Bresch, J., Ban, J., Cnen, D., and Kim, J.: Assimilating AOD retrievals from GOCI and VIIRS to forecast surface PM2.5 episodes over Eastern China, 179, 288–304, https://doi.org/10.1016/j.atmosenv.2018.02.011, 2018.

Pham, D. T., Verron, J., and Gourdeau, L.: Singular evolutive Kalman filters for data assimilation in oceanography, C. R. Acad. Sci. Ser. II, 326, 255–260, https://doi.org/10.1016/S1251-8050(97)86815-2, 1998.

Pires, C., Vautard, R., and Talagrand, O.: On extending the limits of variational assimilation in nonlinear chaotic systems, 48, 96–121, https://doi.org/10.1034/j.1600-0870.1996.00006.x, 1996.

Vetra-Carvalho, S., van Leeuwen, P. J., Nerger, L., Barth, A., Altaf, M. U., Brasseur, P., Kirchgessner, P., and Beckers, J.-M.: State-of-the-art stochastic data assimilation methods for high-dimensional non-Gaussian problems, Tellus A: Dynamic Meteorology and Oceanography, 70, 1–43, https://doi.org/10.1080/16000870.2018.1445364, 2018.

Whitaker, J. S. and Hamill, T. M.: Ensemble data assimilation without perturbed observations, Mon. Wea. Rev., 130, 1913–1924, https://doi.org/10.1175/MWR3156.1, 2002.

Wu, L., Mallet, V., Bocquet, M., and Sportisse, B.: A comparison study of data assimilation algorithms for ozone forecasts, J. Geophys. Res., 113, D20310, https://doi.org/10.1029/2008JD009991, 2008.

Zheng, H.: Improvement of PM2.5 Forecast by Data Assimilation of Ground and Lidar Observation, doctor, University of Science and Technology of China, 2018.

---

## Author Comment (AC5)

Dear Prof. Huang,

Thanks for your effort to review the manuscript and constructive comments concerning our manuscript "An aerosol vertical data assimilation system (NAQPMS-PDAF v1.0): development and application" (MS No. : gmd-2021-374).

**Responses to the comments:**

**Comment:** After checking your manuscript, it has come to our attention that it does not comply with our Code and Data Policy.

https://www.geoscientific-model-development.net/policies/code_and_data_policy.html

GMD can not accept embargoes such as registration or previous contact with the authors to get access to data or code. Therefore, please, to be able of considering your paper for publication you must publish the code and data that you have used for your work in one of the appropriate repositories according to our policy.

We understand that some files used in your study can be large (e.g., full output from models). In such cases, instead of storing the complete files, you should at least keep the variables or final fields computed and used in your work.

Please, when publishing the code, be aware that If you do not include a license, the code continues to be your property and can not be used by others, despite any statement on being free to use. Therefore, when uploading code, you could want to choose a free software/open-source (FLOSS) license. We recommend the GPLv3. You only need to include the file 'https://www.gnu.org/licenses/gpl-3.0.txt' as LICENSE.txt with your code. Also, you can choose other options that Zenodo provides: GPLv2, Apache License, MIT License, etc.

**Reply:**

Thanks for your advice. For the code of NAQPMS-PDAF in our manuscript, we have already set the access as "Open Access" on 31 December 2021 and the source code have been downloaded four times now, which can be found in the following Figure (Figure EC1). The source codes, observation data and model output can be directly downloaded without any access restriction.

We also appreciate the reviewers' comments, which help us to improve the quality of the article. Therefore, in this round of revision, in addition to opening up the source code and data mentioned above, we also upload all the data in the figures and tables in the manuscript into the open-source space together, so that we can share and discuss them easily. We have uploaded all of them to ZENODO (https://doi.org/10.5281/zenodo.6344181) and detailed information can be found in Table EC1.

**Table EC1.** The path of model output and observation data

| | Label | Path |
|---|---|---|
| Figure 1 | Diagram | * |
| Figure 2 | | * |
| Figure 3 | Observation data | ~/Obs |
| Table 1 | Summary | * |
| Figure 4 | Timing information | ~/Output/Timing |
| Figure 5 | | |
| Figure 6 | | |
| Table 2 | Summary | * |
| Figure 7 | Model output | ~/Output/PriorRMSE_TotSpread |
| Figure 8 | Model output + observation data | ~/Output/EXT |
| Figure 9 | | |
| Figure 10 | | ~/Output/PM2.5-STAT |
| Figure 11 | | |
| Figure 12 | | ~/Output/EXT ~Output/CALIPSO |
| Figure 13 | | ~/Output/AERONET |
| Figure 14 | | ~/Output/EXT ~/Obs |
| Figure 15 | Model output | ~/Output/ENS |
| Table S1 | Summary | * |
| Figure S1 | Model output + observation data | ~/Output/EXT |
| Figure S2 | | |
| Figure S3 | Model output | ~/Output/Sensitivity |
| Figure S4 | | |

[Figure]

**Figure EC1.** Webpage screenshot of the source code of NAQPMS-PDAF (http://doi.org/10.5281/zenodo.6344181)